# Attention *(as Discrete-Time Markov)* Chains

**Yotam Erel**[1*]   **Olaf Dünkel**[2*]   **Rishabh Dabral**[2]
**Vladislav Golyanik**[2]   **Christian Theobalt**[2]   **Amit H. Bermano**[1]
[1]Tel Aviv University   [2]MPI for Informatics, SIC
https://yoterel.github.io/attention_chains/

## Abstract

We introduce a new interpretation of the attention matrix as a discrete-time Markov chain. Our interpretation sheds light on common operations involving attention scores such as selection, summation, and averaging in a unified framework. It further extends them by considering indirect attention, propagated through the Markov chain, as opposed to previous studies that only model immediate effects. Our key observation is that tokens linked to semantically similar regions form *metastable* states, i.e., regions where attention tends to concentrate, while noisy attention scores dissipate. Metastable states and their prevalence can be easily computed through simple matrix multiplication and eigenanalysis, respectively. Using these lightweight tools, we demonstrate state-of-the-art zero-shot segmentation. Lastly, we define *TokenRank*—the steady state vector of the Markov chain, which measures global token importance. We show that TokenRank enhances unconditional image generation, improving both quality (IS) and diversity (FID), and can also be incorporated into existing segmentation techniques to improve their performance over existing benchmarks. We believe our framework offers a fresh view of how tokens are being attended in modern visual transformers.

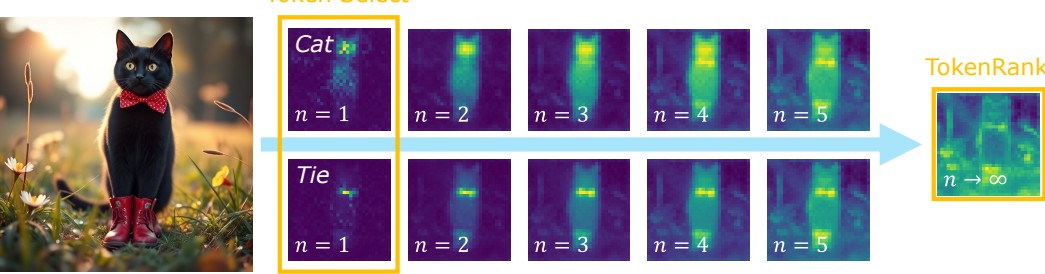

Figure 1: *Attention Chains* interprets attention matrices as Markov chains. The 1[st] order bounce ($n = 1$) corresponds to a common row-select operation from an attention matrix (top: the token "cat", bottom: "tie"). Iteratively computing the n[th] order attention bounce models higher-order attention effects, eventually yielding a stationary vector ($n \to \infty$) that globally captures the flow of attention into each token (*TokenRank*). Intermediate iterations result in sharper segmentation maps.

## 1 Introduction

The attention mechanism (Vaswani et al., 2017) is arguably the most dominant computational primitive in modern deep learning, particularly in the image domain. It has sparked widespread

---

*Equal contribution.

39th Conference on Neural Information Processing Systems (NeurIPS 2025).

research interests and industry adoption, and is extensively leveraged in the literature for anything from segmentation (Khan et al., 2022), through interpretability (Chefer et al., 2021b), to controllable generation (Hertz et al., 2023), in various domains including images (Rombach et al., 2022; Labs, 2024), 3D (Siddiqui et al., 2024; Wu et al., 2024), motion (Dabral et al., 2023; Raab et al., 2024), and more. In the image domain, attention scores play a crucial role: prior studies have leveraged these scores for numerous downstream tasks such as identifying important features (Caron et al., 2021), improved and controlled generation (Hong et al., 2023; Alaluf et al., 2024; Hertz et al., 2023; Tumanyan et al., 2023), visualization and segmentation (Helbling et al., 2025; Dosovitskiy et al., 2021) and analysis of image or video generation models (Hatamizadeh et al., 2024; Wen et al., 2025).

All these examples, however, only examine either partial slices of the attention matrices, or their first few principled components: the most common operations are to select a specific row or column, or to sum up one of the dimensions of the attention matrix that is of quadratic size in sequence length. These operations yield an image-sized map that can be further utilized for the task at hand (e.g., segmentation, generation). Some works integrate information across layers, through direct multiplication (Abnar and Zuidema, 2020; Chefer et al., 2021b), or gradient propagation (Selvaraju et al., 2017; Mehri et al., 2024; Binder et al., 2016), but they still employ the attention operator directly, restricted to capturing only its immediate effects. In contrast, we examine the role of a single attention matrix more holistically, accounting for indirect attention paths via intermediate tokens.

We introduce the interpretation of individual attention matrices as discrete-time Markov chains (DTMC) (Figure 1), where the non-negative attention weights indicate transition probabilities between states that correspond to tokens. This framework, dubbed as *Attention Chains*, allows analyzing token importance through multiple chain transitions, even though the attention operation models only direct relationships. This is analogous to Google's seminal paper on DTMC, PageRank (Page et al., 1999), stating that simple counting of incoming hyperlinks does not generally correspond to page importance. Similarly, simply using direct attention to measure global relevance of a token under performs compared to propagating it through the chain, as we show in various experiments.

We begin by adopting tools from the DTMC literature to explain existing operations commonly applied to attention matrices in a unified framework. These include averaging over heads, selection of rows or columns, and the summation over columns for computing the overall incoming attention for a specific token. Further building on the DTMC interpretation, we extend these operations to new ones. In this regard, our main observation is that tokens in semantically similar regions attend each other, forming a set of *metastable* states (Landim, 2019), where the chain tends to remain for a long time before transitioning elsewhere. In contrast, noisy attention scores tend to disperse rather than cluster.

Based on this insight, we propose a simple yet powerful operation—considering several transitions, or bounces, of the Markov Chain. This consolidates metastable states and helps filter out noise in the attention signal. As we demonstrate, using multiple bounces yields state-of-the-art results for zero-shot segmentation on a commonly used benchmark. Metastable states can also be identified via eigenanalysis, and their presence relates to higher $2^{nd}$ largest eigenvalues. We use this characteristic to weigh attention heads and improve image segmentation.

Lastly, we define *TokenRank*—a concise indicator of the global importance of each token via the steady state vector of the attention Markov chain. TokenRank lends itself well to extracting knowledge encoded in the transformer's attention and improves unconditional image generation, resulting in higher quality and diversity.

We believe that this interpretation is useful for reasoning about the inner workings of modern visual transformers, and can potentially provide a strong foundation for novel applications employing the powerful and widely used attention mechanism.

## 2  Related Work

**Interpretations of Transformer Attention.**   Due the importance of the attention operation in transformers, studies have proposed modeling them using various mathematical frameworks. For example, Ramsauer et al. (2020) relate attention to classic Hopfield networks, Tsai et al. (2019) frame attention as a kernel smoother, Schlag et al. (2021) find its relation to classic fast weight controllers, or Raghu et al. (2021) compare vision transformers to CNNs. El et al. (2025) discuss the mathematical equivalence between message passing in graph neural networks and the self-attention operation in

transformers. Ildiz et al. (2024) consider modeling a 1-layer transformer as a Markov chain. In contrast, we focus on large pretrained transformers and model each individual self-attention matrix as a Markov chain. While their approach allows explaining some typical behaviors of transformers, we explicitly show how our framework allows for the interpretation of existing, pretrained large transformers and for better performance in various downstream tasks. Secondly, we leverage more of the machinery of the Markov chain theory, including multiple transitions and steady state analysis, which was not considered previously.

**Visualization of Attention Maps.** Understanding the attention mechanism of transformers requires probing attention maps across various heads, layers, or timesteps for diffusion models. However, there is no de facto standard for visualizing attention maps. While visualizing cross-attention maps or attention with respect to the class token is common practice (Hertz et al., 2023; Caron et al., 2021), there is no common approach for self-attention due to its many-to-many characteristic. Wen et al. (2025), for example, analyze raw attention maps whereas Hatamizadeh et al. (2024) visualize where attention is flowing into with respect to an image token in the center. Alternatively, several studies visualized the principle components using SVD decomposition (Liu et al., 2024; Tumanyan et al., 2023). On the other hand, we propose extracting a unique steady-state vector of the attention matrix that allows to more holistically visualize where attention is flowing to, or where it is flowing from.

**Explainability for Transformers.** One important branch of explainability research for convolutional and feed-forward networks are gradient-based methods that compute a model's explanation taking into account the gradient of a model prediction with respect to the input pixels (Erhan et al., 2009; Selvaraju et al., 2017; Simonyan et al., 2013). Layer-wise Relevance Propagation (LRP) (Binder et al., 2016) propagates relevance scores backward through the neural network, which has been extended to transformers (Chefer et al., 2021b,a; Achtibat et al., 2024). A different line of work makes use of the attention mechanism to propagate where attention is flowing through the network (Abnar and Zuidema, 2020) or combines it with a gradient-based strategy (Bousselham et al., 2024). Our framework focuses on individual attention matrices, and naturally allows to better explain each attention matrix by considering indirect interaction effects. This is perpendicular to studies that attempt to explain the input using the output in an end-to-end fashion.

**Extracting Knowledge Encoded in Foundational Models.** Recently, larger interest was raised in how to extract knowledge encoded in foundational models (e.g. DINO (Caron et al., 2021; Oquab et al., 2024), CLIP (Radford et al., 2021), or generative models such as FLUX (Labs, 2024)) and to better understand their inner workings. Caron et al. (2021); Chefer et al. (2021b); Hao et al. (2021); Tang et al. (2023) use raw attention maps for computing attribution scores, while Gandelsman et al. (2024) investigate the CLIP image encoder by decomposing the image representation and relating it to CLIP's text representation. Helbling et al. (2025) propose the usage of concept tokens that allow higher quality extraction of text-token specific attention maps even if they do not appear in the prompt. Other recent works focus on extracting noise-free diffusion features (Stracke et al., 2025), reducing massive attention activations (Gan et al., 2025), or refining features with a light-weight adapter (Dünkel et al., 2025). Nguyen et al. (2023) perform self-attention exponentiation for refining semantic segmentation maps. We explore how propagating indirect attention effects through a Markov chain helps in extracting more informative signals for various downstream tasks.

## 3 Preliminaries

### 3.1 Discrete-Time Markov Chains (DTMC)

A right-stochastic square matrix $\mathbf{A} \in \mathbb{R}^{n \times n}$ where all rows sum up to one $\sum_j \mathbf{A}_{i,j} = 1, \quad \forall i$, and with non-negative entries $\mathbf{A}_{i,j} \geq 0, \quad \forall i,j$ is a transition probability matrix of a time-homogeneous DTMC with $n$ states, where $\mathbf{A}_{i,j} = P(j|i)$ is the probability of transitioning from state $i$ to state $j$. If the directed weighted graph represented by the transition matrix is irreducible and aperiodic, then there exists a unique eigenvector $\mathbf{v}_{ss}$ corresponding to the eigenvalue 1: $\mathbf{A}^T \cdot \mathbf{v}_{ss} = 1 \cdot \mathbf{v}_{ss}$. This eigenvector is stationary and describes the steady state that any given initial state of the system evolves into over time. Another useful property of the transition matrix $\mathbf{A}$ is that the size of its second largest eigenvalue $|\lambda_2|$ indicates how slowly the chain converges into $\mathbf{v}_{ss}$, or alternatively, the number of metastable states in the chain.

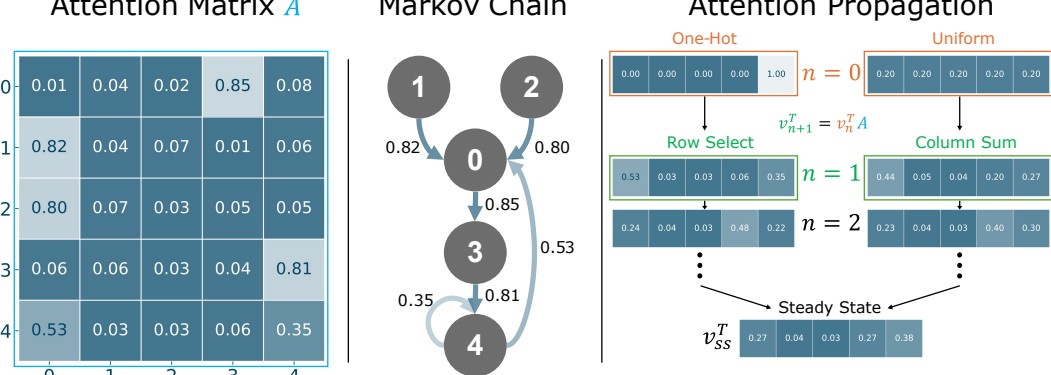

Figure 2: **Illustration of higher order effects. Left:** Attention matrix $A$ with sequence length 5. **Middle:** A DTMC with transition probabilities defined by matrix $A$, where only strong connections are shown. **Right (One-Hot):** To evaluate where *state-4* attends to, we can iterate using the power method once starting from a one-hot vector ($n = 0$), which results in the row-select operation ($n = 1$). However, this first-order approximation is insufficient since state-0 mostly transitions to *state-3* and, therefore, *state-4* indirectly attends state-3. This becomes evident as we iterate further ($n = 2$). **Right (Uniform):** To compute a global token ranking, we can iterate starting from a uniform state ($n = 0$), resulting in a per-column sum operation ($n = 1$). This indicates *state-0* as most important because many states have a high probability of transitioning into *state-0*. However, *state-0* maps to *state-3* with high probability, and *state-3* maps to *state-4* with high probability. Therefore, the importance of *state-4* should be elevated. When considering the second bounce ($n = 2$), more probability mass is directed into *state-3*, and with a sufficient number of iterations the steady state ($v_{ss}^T$) ranks *state-4* as the most important state globally, which aligns with the intuition above.

**Power Method.** The most straightforward way to obtain the steady-state vector $\mathbf{v}_{ss}$ is using the *power method* that iteratively computes

$$\mathbf{v}_{n+1}^T = \mathbf{v}_n^T \mathbf{A}, \tag{1}$$

where $\mathbf{v}_{n=0}^T$ is an initial state summing up to one. The computation is usually terminated once the norm falls below a specific threshold $||\mathbf{v}_{n+1}^T - \mathbf{v}_n^T||_2^2 < \tau$ or after some fixed number of iterations.

### 3.2 PageRank Algorithm

As we rely on their formulation throughout this paper, we briefly discuss Google's PageRank (Page et al., 1999) algorithm. PageRank models the hyperlink structure of the web as a stochastic process $\mathbf{P}$ where states are web pages, and outgoing links are used to define transition probabilities. Here, the matrix $\mathbf{P}$ is row normalized and pages with no outgoing links are replaced with uniform vectors. To ensure a unique steady state vector, the considered matrix needs to be irreducible and primitive. For this purpose, a revised transition matrix $\mathbf{P}'$ with only positive entries is computed from $\mathbf{P}$:

$$\mathbf{P}' = \alpha \mathbf{P} + (1 - \alpha) \frac{1}{n} \mathbf{e}\mathbf{e}^T, \tag{2}$$

where $0 < \alpha < 1$ is a hyper parameter controlling the transitioning probability of "teleportation" into a random state, and $\mathbf{e}$ is a column vector of all ones. Applying the power method on $\mathbf{P}'$ results in a unique vector $\vec{\pi}$, the global PageRank vector, inducing the order $i \preceq_{\vec{\pi}} j \iff \vec{\pi}_i \leq \vec{\pi}_j$.

## 4 Attention as a Discrete-Time Markov Chain

After softmax, the attention matrix becomes a right-stochastic matrix with non-negative entries:

$$\mathbf{A} = \text{softmax}\left(\frac{\mathbf{Q}\mathbf{K}^T}{\sqrt{d_h}}\right) \in \mathbb{R}^{h \times s_1 \times s_2}, \tag{3}$$

where $h$ is the number of heads, $s_{1,2}$ are the token sequence lengths, and $d_h$ is a sub-space of the full embedding dimension $d$. Consequently, we interpret the attention matrix as a DTMC, going

beyond the conventional attention mechanism in transformers, which only models direct interaction between queries and keys. Our discussion is limited to attention blocks of equal sequence size (e.g. self-attention, or hybrid attention blocks), unless otherwise stated. We start by formalizing some common operations performed on attention matrices using the DTMC formalism (Section 4.1). While the previously proposed attention operations can be formulated with a single bounce of the Markov chain, we go beyond this and iterate through the Markov chain. Specifically, we describe three new operations enabled by this framework: multi-bounce attention (Section 4.2); where higher-order attention effects are taken into account, *TokenRank* (Section 4.3); the token-space equivalent of PageRank, and $\lambda_2$ weighting; an improved attention head weighting scheme that is based on the second largest eigenvalue of the chain.

## 4.1 Interpretation of Existing Attention Operations

In this section, we show that common operations on attention matrices—for purposes of visualization, explainability, and information extraction—can be interpreted within the DTMC framework. We use $\mathbf{A}$ to denote an attention matrix after the softmax operation.

**Multiplying** attention matrices of subsequent blocks of a transformer is performed by previous explainability studies (Dosovitskiy et al., 2021; Abnar and Zuidema, 2020; Chefer et al., 2021b) that propagate information from the output to the input. This operation is is equivalent to chaining several Markov chains in the DTMC framework. Another way to view this is a time-dependent Markov chain, where in each time step $t$ the next matrix in the multiplication order represents transition probabilities.

**Adding** an identity matrix to an attention matrix and re-normalizing it was used in previous studies to mitigate the effects of skip connections (Abnar and Zuidema, 2020; Chefer et al., 2021b). This is equivalent to computing a new chain $\mathbf{A}' = 0.5(\mathbf{I} + \mathbf{A})$. Then, the following holds:

$$\mathbf{A}'\mathbf{v}_{ss} = 0.5(\mathbf{I} + \mathbf{A})\mathbf{v}_{ss} = 0.5(\mathbf{v}_{ss} + \mathbf{A}\mathbf{v}_{ss}) = 0.5(\mathbf{v}_{ss} + \lambda\mathbf{v}_{ss}) = 0.5(1 + \lambda)\mathbf{v}_{ss} = \mathbf{v}_{ss},$$

where $\mathbf{v}_{ss}$ is a the steady-state vector of $\mathbf{A}$ and the last transition is because the eigenvalue associated with the steady state is $\lambda = 1$. Evidently, this operation does not change the system's steady state, preserving its global nature. Instead, the chain converges more slowly since each state has a higher transition probability to itself.

**Selecting** a specific **row** $\mathbf{v}_i^T = (\mathbf{A}_{i,j})_{j=1}^n$, corresponding to a single token $i$ results in a row-vector of attention given by token $i$ to all other tokens, in a one-to-many relationship. In our framework, $\mathbf{v}_i^T$ can be obtained by performing a single state transition $\mathbf{v}_i^T = \mathbf{u}_i^T\mathbf{A}$, where $\mathbf{u}_i^T$ is a one-hot row vector on index $i$.

**Selecting** a specific **column** $\mathbf{v}_j = (\mathbf{A}_{i,j})_{i=1}^n$, corresponding to a single token $j$, encodes which other tokens find $j$ important. This operation is commonly used to visualize a many-to-one relationship, such as image tokens attending specific text tokens in a cross-attention matrix (Dosovitskiy et al., 2021; Hertz et al., 2023). Since column-select operates in column-space (as opposed to row-select), which is not a valid probability distribution, we first normalize the columns to acquire a left-stochastic matrix $\mathbf{A}_l$. This allows us to equate a column-select operation up to some constant scale as: $\mathbf{v}_j \propto \mathbf{A}_l\mathbf{u}_j$, where $\mathbf{u}_j$ is a one-hot column vector for index $j$.

Thus, our formulation allows us to reinterpret the selection operation as the first bounce transition from a one-hot vector, a view that we will further motivate in Section 4.2.

**Summing** each column of $\mathbf{A}$ results in a row vector $\mathbf{v_s}^T$ that aggregates attention given to each token from all other tokens. In our formulation, $\mathbf{v_s}^T$ can be obtained by performing a single state transition: $\mathbf{v}_s^T = \frac{1}{n}\mathbf{e}^T\mathbf{A}$, where $\mathbf{e}^T$ is a row vector of ones. This operation can be a useful first-order approximation in ranking tokens' importance globally, as it equates to starting from an equilibrium state and transitioning through the DTMC process once. However, we show this straight-forward approach under-performs compared to the global TokenRank vector (Section 4.3) that takes into account indirect attention being given to all tokens. See Figure 2 for an illustrative example.

**Averaging** multiple attention matrices over different attention heads is a common approach to process or visualize the information contained in attention layers of transformers. This is despite numerous studies indicating that some heads are not informative for large transformers (Voita et al., 2019; Chefer et al., 2021b), suggesting that simple averaging might dilute the signal under observation. In the DTMC framework, averaging yields a new process where, at each step, the transitioning probabilities are the mean of those from the original chains — a "mixture" of the chains with equal

weighting. This operation can be reasonable if the chains are not very different. As an alternative, we propose a weighting scheme that considers the convergence rate of the chains in Section 4.4.

## 4.2 Multi-Bounce Attention

When the subject of interest is a specific token $i$, the core shortcoming with row- and column-select from a Markovian perspective is that they do not take into account higher order effects: As illustrated in Figure 2, the tokens that token $i$ attends to are also attending other tokens, indirectly affecting token $i$. While it is evident that the attention mechanism only implements direct influence between tokens, we show that considering indirect paths allows for a deeper understanding of attention flow within individual attention heads. For brevity, we will now extend the row-select operation using our framework, which can be analogously applied to column-select after normalizing the columns of $\mathbf{A}$ and transposing.

The crucial observation enabled by the DTMC interpretation is that iterating on the selection operation $n$ times for token $i$ yields

$$\mathbf{v}^T_{\{i,n+1\}} = \mathbf{v}^T_{\{i,n\}}\mathbf{A}, \tag{4}$$

with $\mathbf{v}^T_{\{i,n=0\}}$ being a one hot row vector. Such an iterative process is mathematically equivalent to the power method (Equation (1)). In other words, if we wish to take into account the $n^{\text{th}}$ order attention bounce, we can simply apply this formula $n$ times. For a single bounce, $\mathbf{v}^T_{\{i,n=1\}}$ captures incoming attention for tokens that $i$ attends to directly (row-select) or outgoing attention for tokens that attend $i$ directly (column-select). For any $\mathbf{v}^T_{\{i,n>1\}}$, higher order bounces are considered. See Figure 1 for an illustration.

## 4.3 TokenRank

In many cases, a more complete understanding of token importance is required. Observe that applying multi-bounce attention (Section 4.2) will converge into a stationary vector *for any initial state* if specific conditions are met (Section 3.1). In the context of attention matrices, we name this vector *TokenRank*, inspired by Google's webpage stationary vector PageRank (Page et al., 1999). We argue that TokenRank is a better tool to measure the global importance of tokens in a single attention head, compared to more localized and naïve attempts to probe the attention matrix (e.g. by selecting a specific column, or summing up each of the columns).

One caveat is that attention weight matrices might still be reducible, or have cycles of fixed length. This means they do not necessarily hold the properties that guarantee the existence and uniqueness of a steady-state vector. A sufficient way to guarantee this, is to adjust $\mathbf{A}$ using the PageRank formulation (Equation (2)) prior to iterating.

While this formulation allows us to immediately acquire authoritative tokens (those for which attention is flowing into), we can also normalize the columns followed by transposing of $\mathbf{A}$. Then, iterating obtains important hub tokens (those which attention is flowing out from), obtaining two TokenRanks: one for incoming and one for outgoing attention.

We show that TokenRank extracts a cleaner signal of the attention (Section 5.2), can be used to improve downstream tasks (Section 5.3), and is better for determining the importance of tokens in the sequence compared to other approaches (Section 5.5).

## 4.4 $\lambda_2$ Weighting

The mixing time of a DTMC into its steady state is mathematically tied with its $2^{\text{nd}}$ largest eigenvalue $\lambda_2$ (Langville and Meyer, 2004). Intuitively, larger $\lambda_2$ values correspond to more stable metastable states, indicating that more important information is captured by the attention matrix. Indeed, DTMCs with randomly sampled transition probabilities tend to have very small $\lambda_2$ (see Appendix F).

To this end, instead of simply averaging attention matrices over the head dimension (Section 4.1), we instead propose to perform weighted averaging over the heads using their $\lambda_2$ values. We show this weighting scheme results in better downstream task performance (see Section 5.1 and Appendix E.2).

| Orig. Img. | Colum Select | ConceptAttention | Ours | GT Mask |
|---|---|---|---|---|

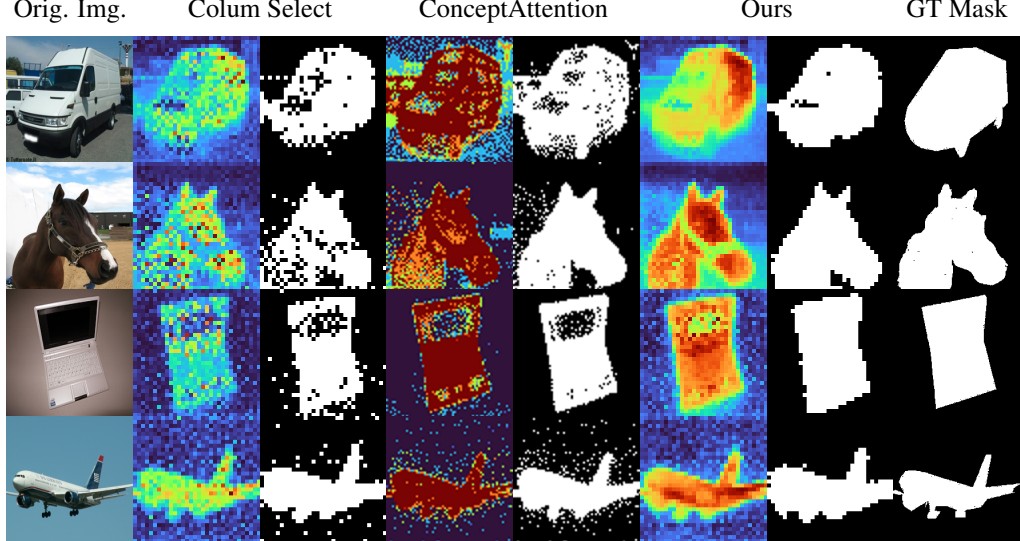

Figure 3: **ImageNet segmentation.** Considering higher order attention effects improves results. We visualize the raw attention output (colored) and the binary segmentation masks. We present more qualitative comparisons in Appendix C.2 .

# 5 Experiments

This section provides extensive evidence that our framework and the extended operations over the attention matrices are useful for a variety of downstream tasks. We start with improving zero-shot segmentation over a standard benchmark using multi-bounce attention (Section 5.1), followed by showing that TokenRank extracts a more informative signal than existing baselines (Section 5.2), useful for visualization purposes. Then, we demonstrate that applying TokenRank to existing techniques improves unconditional generation of images (Section 5.3) and segmentation (Section 5.4). This is supplemented with a token masking experiment (Section 5.5), where we deliberately mask tokens according to their TokenRank importance and measure downstream accuracy for image classification.

## 5.1 ImageNet Zero-Shot Segmentation

A straightforward way to evaluate the usefulness of taking into account higher-order attention effects is to measure the performance on a downstream task such as zero-shot segmentation. We evaluate on the ImageNet Segmentation benchmark (Chefer et al., 2021b) consisting of 4276 images with 445 categories and we employ FLUX-Schnell (Labs, 2024), a pretrained transformer-based generative model, to denoise the images and extract the attention maps. We then average the maps over the heads using $\lambda_2$ weighting (Section 4.4) and compute the multi-bounce attention when considering outgoing links with $n = 2$ (Section 4.2). We refer to the Appendix A for further implementation details and to Appendix C for further results.

We report the results for the current state-of-the-art method (Helbling et al., 2025) and additionally provide the row- and column-select operations as baselines using their implementation. The results are reported in Table 1 and example qualitative segmentation masks are presented in Figure 3, indicating that our method is better than the previous state-of-the-art, as the threshold-agnostic metric mAP shows a large gap in performance. We hypothesize that this is because semantically similar regions form sets of metastable states, and iterating consolidates them, which results in cleaner segmentation maps. Our proposed $\lambda_2$ weighting further improves the segmentation performance, as we show in statistical tests in Appendix E.1.

Table 1: **ImageNet segmentation.** Our method yields state-of-the-art results. Error bars and further results are reported in Appendix C.

| Method | Architecture | Acc ↑ | mIoU ↑ | mAP ↑ |
|---|---|---|---|---|
| LRP (Binder et al., 2016) | ViT-B/16 | 51.09 | 32.89 | 55.68 |
| Partial-LRP (Binder et al., 2016) | ViT-B/16 | 76.31 | 57.94 | 84.67 |
| Rollout (Abnar and Zuidema, 2020) | ViT-B/16 | 73.54 | 55.42 | 84.76 |
| ViT Attention (Dosovitskiy et al., 2021) | ViT-B/16 | 67.84 | 46.37 | 80.24 |
| GradCam (Selvaraju et al., 2017) | ViT-B/16 | 64.44 | 40.82 | 71.60 |
| DiffSeg Tian et al. (2024) | SD1.4 | 65.41 | 52.12 | - |
| TextSpan (Gandelsman et al., 2024) | ViT-H/14 | 75.21 | 54.50 | 81.61 |
| TransInterp (Chefer et al., 2021b) | ViT-B/16 | 79.70 | 61.95 | 86.03 |
| DINO Attention (Caron et al., 2021) | ViT-S/8 | 81.97 | 69.44 | 86.12 |
| DAAM (Tang et al., 2023) | SDXL UNet | 78.47 | 64.56 | 88.79 |
| FLUX Cross Attention (Helbling et al., 2025) | FLUX DiT | 74.92 | 59.90 | 87.23 |
| FLUX row-select | FLUX DiT | 73.96 | 54.65 | 82.64 |
| FLUX column-select | FLUX DiT | 80.55 | 64.02 | 87.20 |
| Concept Attention (Helbling et al., 2025) | FLUX DiT | 83.07 | **71.04** | 90.45 |
| Ours w/o $\lambda_2$ | FLUX DiT | 84.00 | 70.02 | 94.28 |
| Ours | FLUX DiT | **84.12** | 70.20 | **94.29** |

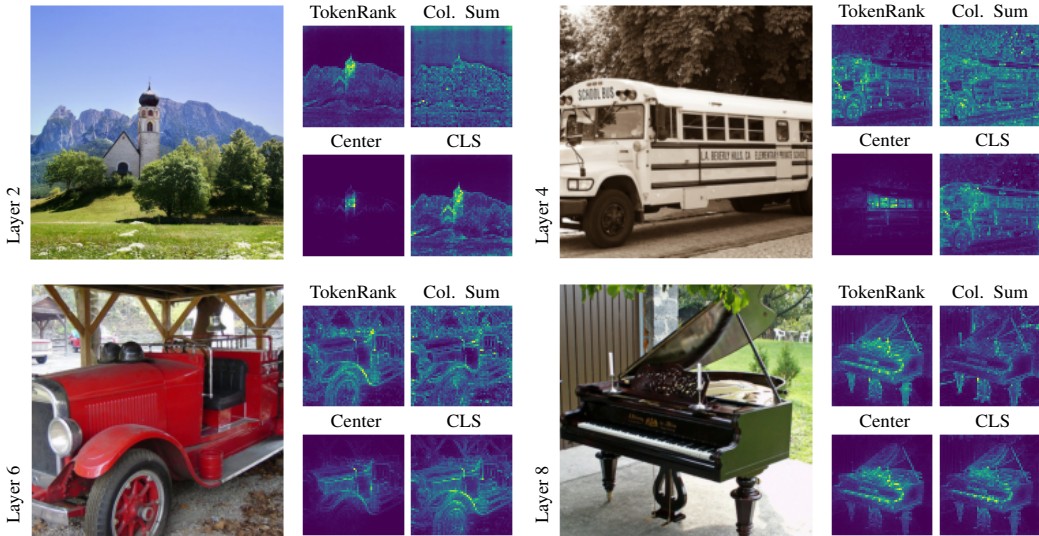

Figure 4: **Global incoming attention.** Visualizations are computed after averaging over heads for four different layers of DINOv1. While the center token only attends to the local neighborhood for earlier layers, column sum results in noisy attention visualizations. In contrast, TokenRank captures global incoming attention on par with the CLS token that was explicitly trained to capture global attention for DINOv1. We show per-head visualizations in Appendix H.

## 5.2 TokenRank for Visualizing Attention Maps

Previous typical approaches for visualizing global incoming attention consider the attention with respect to a *Center Token*, that is, row-select for that token (Hatamizadeh et al., 2024), or compute a per-*Column Sum* (Hong et al., 2023). These methods share the common characteristic that they can each be expressed as a single power-method iteration (Section 4.1). In other words, they only take into account direct attention effects. On the other hand, TokenRank considers indirect attention paths to evaluate where attention is flowing into. This results in attention maps that are sharper and less noisy than with column sum and more complete than using a single token (Figure 4). We provide a quantitative comparison through a linear probing experiment Appendix B.

## 5.3 Improving Self-Attention Guidance (SAG)

To evaluate TokenRank on a downstream task, we integrate it in SAG (Hong et al., 2023), an approach that improves the fidelity and the quality of unconditional image generation with denoising diffusion models. SAG uses the self-attention weight matrices to form a mask that indicates important spatial tokens. It then adversarially blurs the masked regions and drives the denoising process away from it, creating better images as a result. To find out where important features exist, Hong et al. (2023) average over the head dimension for a specific attention layer, followed by a summing each of the columns (Section 4.1). We compare the quality and diversity of generated images when using the TokenRank instead of their proposed Column-sum strategy. We compute the IS (Salimans et al., 2016), FID (Heusel et al., 2017), and KID (Bińkowski et al., 2018) metrics over 50K generated images. The selected metrics offer complementary information in terms of quality, diversity, and resemblance to ground-truth distribution. Results are reported in Table 2 and Figure 5. We refer to the supplementary material for further details and results. Using TokenRank improves the generated images significantly in both quality and diversity compared to the original SAG, showcasing TokenRank's ability to rank globally important tokens in the sequence, and improve a downstream generative task.

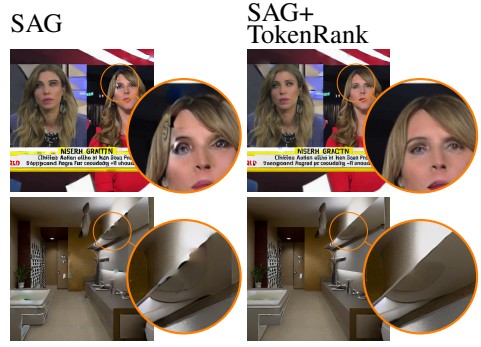

SAG    SAG+ TokenRank

Table 2: **Quantitative results for SAG.** Comparisons between SD1.5 (Rombach et al., 2022), SAG (Hong et al., 2023) and SAG+TokenRank. Metrics were computed over 50K examples.

| Method | IS ↑ | FID ↓ | KID ↓ |
|---|---|---|---|
| SD1.5 | 16.32 | 45.77 | 0.018 |
| SAG | 17.69 | 52.48 | 0.023 |
| SAG+TokenRank | **18.37** | **50.14** | **0.021** |

Figure 5: **Qualitative results for SAG.** Images generated using TokenRank have less artifacts and are more structured.

## 5.4 Improving DiffSeg with TokenRank

We incorporate TokenRank into DiffSeg (Tian et al., 2024), an existing segmentation approach that merges self-attention matrices using KL-divergence to measure similarity between up-sampled attention maps. Specifically, in the original study the authors used a uniform grid to initialize the anchors used as seeds for their proposal algorithm. Instead, we sample anchors according to their TokenRank importance, with suppression to avoid repeatedly selecting the same location.

Table 3: **COCO-Stuff-27 benchmark.** Using TokenRank to sample anchors for initial proposals in DiffSeg (Tian et al., 2024) results in significant improvements over the benchmark.

| Method | mACC ↑ | mIoU ↑ |
|---|---|---|
| Uniform Grid | 72.50 | 43.60 |
| TokenRank Grid | **84.97** | **44.87** |

This better grid strategy leads to substantial improvements over the original approach using the COCO-Stuff-27 benchmark as can be seen in Table 3. This experiment further shows our framework is orthogonal to other solutions leveraging self-attention for solving downstream tasks.

## 5.5 Masking Most Influential Tokens

Inspired by standard faithfulness experiments (Mehri et al., 2024), where changes in the performance of a downstream classifier are observed through progressively occluding input features, we designed a new experiment that targets evaluating strategies that measure the global importance of tokens per individual attention head. Specifically, the most influential tokens in the sequence are masked out progressively by zeroing the corresponding columns in the attention matrices and the resulting classification accuracy drop is measured. We evaluate the accuracy degradation of several vision transformer-based classifiers over 5000 randomly selected images from ImageNet with a fixed seed. To determine the order of tokens to be masked, we use the TokenRank vector and compare it to the

following baselines: randomly selecting a token ("Rand. Token"), using the column of the token corresponding to the center patch of the image ("Center Token"), the per column-sum ("Column Sum"), and using the CLS Token. Note that we do not mask global tokens, i.e., the CLS token and the registers for DINOv2, as this results in a massive drop in performance rendering the comparisons less useful. We refer to Appendix A for further details.

Results are reported in Table 4, illustrating that classification accuracy degrades faster using TokenRank than other baselines for several pre-trained transformers. This validates the hypothesis that TokenRank captures important global information more faithfully since it also models higher-order effects. Another interesting observation is that transformers with a more structured features (e.g. DINOv2 with registers (Darcet et al., 2024)) tend to benefit more from TokenRank, which suggests it can also be potentially used for quantifying this aspect of a trained transformer. We hypothesize that this

Table 4: **Results for masking most influential tokens.** We present the area under curve (AUC) metric for the accuracy normalized by the original model accuracy and average over various models of a selected model family with model-specific results presented in Appendix D. Using TokenRank to mask out important tokens consistently results in larger accuracy drops than other strategies.

| AUC ↓ | ViT | CLIP | DINOv1 | DINOv2 |
|---|---|---|---|---|
| Rand. Token | 0.79 | 0.80 | 0.88 | 0.89 |
| Center Token | 0.33 | 0.47 | 0.45 | 0.70 |
| Column Sum | 0.27 | 0.49 | 0.47 | 0.71 |
| CLS Token | 0.33 | 0.53 | 0.56 | 0.70 |
| TokenRank | **0.26** | **0.46** | **0.44** | **0.64** |

is due to TokenRank's tendency to sharpen the input signal (Figure 4), potentially reinforcing the effect of noise or artifacts for an unstructured feature space.

## 6 Conclusion

In this paper, we proposed a novel interpretation of the attention map as a DTMC. By treating such maps as stochastic matrices, we showed that taking into account higher-order interactions between tokens results in better performance on various downstream tasks. We further showed the stationary vector TokenRank can help depict global incoming and outgoing attention within a single head.

**Limitations and future work.** Our approach is limited to square matrices (e.g. self-attention / hybrid-attention blocks). Therefore, cross-attention blocks do not naturally fit into our Markov chain formulation due to the existence of non-accessible states. There could potentially be a way to resolve this by introducing dummy states with uniformly distributed transition probability. Additionally, computing the second eigenvalues $\lambda_2$ is computationally heavy. Computing multiple bounces or the TokenRank on the other hand is quite performant, and converges typically after 10 to 20 iterations. Finally, we observe that the performance gains with TokenRank are larger for transformers where the attention map is more structured. We envision that this framework opens up new possibilities for analyzing, understanding, and modeling the attention operation, and may be used in conjunction with other end-to-end explainability tools to enhance understanding of visual transformers. This can also serve various other domains such as video, audio, motion, as well as natural language processing.

## Acknowledgments

This research was supported in part by the Len Blavatnik and the Blavatnik family foundation and ISF number 1337/22. We would like to thank Hila Chefer and Jonas Fischer for the valuable feedback. We further thank Siddhartha Gairola, Amin Parchami, and Wanyue Zhang for fruitful discussions.

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

# Attention (as Discrete-Time Markov) Chains

## Supplementary Material

## A  Implementation Details

### A.1  ImageNet Zero-Shot Segmentation

To establish fair comparisons, we followed the exact procedure described by Helbling et al. (2025) using the code from `https://github.com/helblazer811/ConceptAttention`. We simplify all categories to a single word, allowing methods conditioned on text to select the corresponding token. We use the timestep-distilled Flux-Schnell DiT (Labs, 2024) and add Gaussian noise to each of the images to the normalized timestep $t = 0.5$, resulting in two backward steps for denoising the image. We use the last 10 layers of the multi-stream attention blocks. The attention matrices from all aforementioned layers and timesteps are averaged. For the heads, we use the $\lambda_2$ weighting for averaging. For multi-bounce attention, we use the column-select formulation, i.e. we normalize the columns and transpose $A$ as described in Section 4.2 of the paper.

### A.2  Improving Self Attention Guidance (SAG)

For the SAG experiments, we build on top of the code of (Hong et al., 2023) provided in `https://github.com/SusungHong/Self-Attention-Guidance`. We use SD1.5 (Rombach et al., 2022) as the foundational model and use the second last decoder block's last self-attention layer for masking. We empirically observed it consistently resulted in the best results for both SAG and SAG+TokenRank, as opposed to the usage of the bottleneck layer in the comparison experiments in the original SAG, which is in line with the observation by Hong et al. (2023).

We compute the IS (Salimans et al., 2016), FID (Heusel et al., 2017), and KID (Bińkowski et al., 2018) metrics over 50K generated images, where the ground truth dataset is the LAION-Aesthetics V2 dataset (Schuhmann et al., 2022) - originally used to train SD1.5. We used the $score > 6.5$ subset consisting of 625K images, and filtered it for $score > 6.65$ and images with width and height larger than 512, resulting in 72495 raw links, out of which roughly 50K were valid. After truncating to 50K, these images were used for computing the metrics. The selected metrics offer complementary information in terms of quality (IS) and diversity and resemblance to ground-truth distribution (FID / KID).

We use the unconditional branch only, and allow up to 40% of the attention matrix to be masked, following the original implementation.

### A.3  Improving DiffSeg with TokenRank

We used the official code of Tian et al. (2024) provided in `https://github.com/google/diffseg`. DiffSeg aggregates attention weights from all layers, where deeper layers are up-sampled. The aggregation is also a Markov Chain (see Section 4.1) and, therefore, TokenRank can be computed once per image. To create the TokenRank grid, we simply compute TokenRank and sort it by descending order, and greedily select the highest ranked token that is at least 5 units away from all currently selected tokens to act as suppression which avoid oversampling the same region. If not enough anchors can be selected in this way, we remove the suppression and select the rest of the tokens by only considering their TokenRank. We use the same total amount of anchors as the baseline for fairness.

### A.4  Masking Most Influential Tokens

For all transformers, we do not allow masking the CLS token in our evaluations, as it consistently ranks as the top most important token for almost all heads and layers, and discarding it yields an extreme drop in accuracy, which does not allow for comparisons of the alternative approaches. For the same reason, we also only use the first half of the available attention layers for each model. This allows for not disturbing the output space too much and as a result accuracy performance degrades gracefully. Refer to Appendix D.2 for further justification of this choice. Moreover, we mask out

Table 5: **Linear probing.** TokenRank extracts a more informative signal that captures global incoming attention and results in a larger classification accuracy on the *Imagenette* dataset.

| Accuracy | DINOv1 | DINOv2 | CLIP |
|---|---|---|---|
| Rand. Token | $67.33 \pm 3.12$ | $63.32 \pm 6.04$ | $57.65 \pm 2.66$ |
| Center Token | $66.32 \pm 3.08$ | $75.34 \pm 1.85$ | $68.42 \pm 2.66$ |
| Column Sum | $75.64 \pm 2.74$ | $90.76 \pm 2.21$ | $73.73 \pm 2.98$ |
| TokenRank | $\mathbf{77.31 \pm 2.50}$ | $\mathbf{92.73 \pm 1.51}$ | $\mathbf{73.88 \pm 2.86}$ |
| CLS Token | $81.53 \pm 2.44$ | $94.07 \pm 0.97$ | $72.46 \pm 3.40$ |

the same number of tokens for all heads simultaneously. For DINO family type, results in the main paper are averaged over the ViT-B/16, ViT-B/32, ViT-S/16, and ViT-S/32 architectures. For DINOv2 family type, we use the w/ registers variation, and average over the ViT-B/14, ViT-S/14 and ViT-L/14 architectures. For both DINO and DINOv2 family types, we use pretrained linear classifier heads provided with the original publications. We average ViT/B and ViT/L for the transformer models trained in a supervised way. For CLIP family type, we average over the ViT-B/32, ViT-L/14 and ViT-L/14/336 architectures. We use the template prompt "A photo of a <class>" to compute per-class text features. Then, we compute a dot-product with the image features and these per-class features to determine the classification result. Masking is performed by setting the entire column corresponding to a token to $-\infty$ prior to softmax. For a more in-depth view of individual model accuracy degradation, see Appendix D.

## B    Linear Probing of TokenRank

To quantitatively evaluate that TokenRank produces sharper maps and captures the global information contained in the attention matrices, we train a linear classifier on top of all proposed attention visualizations, inspired by a previous similar analysis (Liu et al., 2024). For this purpose, we extract attention visualizations for all layers, heads, and images for the ImageNet (Deng et al., 2009) subset *Imagenette* (Howard, 2019) for three foundation models without considering the CLS token. The results in Table 5 show that TokenRank visualizations result in a cleaner signal for image classification than previous approaches that only consider one bounce of attention. It is on par with row-selecting the CLS token, which is an upper bound because it was specifically trained to capture class concepts for the DINO model family. Note error bars are reported over 5 different runs each with a different seed. We performed a Wilcoxon test with the null hypothesis that the competing column sum operation results in the same accuracy as TokenRank and the one-sided alternative hypothesis that the accuracy is lower. We reject the null hypothesis for DINOv1 with p=0.03 and for DINOv2 with p=0.03.

We generate attention maps for all layers, heads, and images for the ImageNet (Deng et al., 2009) subset *Imagenette* (Howard, 2019) for DINOv1, DINOv2, and CLIP with a VIT/B model. Then, we compute attention visualizations with column sum, TokenRank, row-selecting the center token, and row-selecting the CLS token. Finally, we train a linear classifier with the commonly used train and test split and with the attention visualizations of all layers and heads as input, optimized with the SGD optimizer with a set of learning rates ($\{1e-5, 2e-5, 5e-5, 1e-4, 2e-4, 5e-4, 1e-3\}$)) for 20 epochs, and we choose the best performing model, similar to Oquab et al. (2024). We resize and center crop the images to get $905 \times 905$ ($901 \times 901$) attention maps for DINOv2 with registers (other models). We exclude the CLS token column for classification.

**Linear Probing of $\lambda_2$-Weighted Aggregated Heads**    Additionally, we also train linear classifiers on top of the TokenRank vectors (incoming attention) per layer, where we use different weighting schemes for the head dimension. For this, we first compute the weighting coefficients per head and perform a weighted average of the heads. Then, we compute TokenRank for the aggregated results. We compare uniform weighting (simple average), random weighting, and $\lambda_2$ weighting. Results are presented in Table 6: While random weighting is significantly worse, $\lambda_2$ weighting performs better than simple averaging, particularly for transformers with a well-structured latent space, such as DINOv2 with registers.

Table 6: **Linear probing of steady state vectors after aggregating heads.** $\lambda_2$ weighting is better than uniform weighting.

| Method | uniform | random | $\lambda_2$ |
|--------|---------|--------|-------------|
| CLIP   | **49.07** | 44.07 | **49.07** |
| DINOv1 | 53.50   | 47.21 | **53.55** |
| DINOv2 | 71.62   | 50.29 | **72.36** |

Table 7: **ImageNet segmentation.** Ablation and hyper-parameter choice.

| Method | Acc $\uparrow$ | mIoU $\uparrow$ | mAP $\uparrow$ |
|--------|------|------|------|
| row-select ($n = 1$) | 73.96 | 54.65 | 82.64 |
| column-select ($n = 1$) | 80.55 | 64.02 | 87.20 |
| single-channel | 69.63 | 51.73 | 80.50 |
| incoming ($n = 2$) | 83.73 | 69.58 | 93.68 |
| incoming ($n = 3$) | 82.93 | 68.81 | 94.01 |
| outgoing ($n = 3$) | 79.25 | 63.32 | 89.80 |
| outgoing ($n = 2$)* | **84.12** | **70.20** | **94.29** |

## C    Additional Segmentation Results

### C.1    Ablation and Hyperparameters

See Table 7 for zero-shot ImageNet segmentation results using the same experimental setup as in the main paper, with different hyper parameter choices. The bottom of Table 7 shows "outgoing ($n = 2$)*", corresponding to the result shown in the main paper, leveraging the second bounce from a one-hot column vector and utilizing $\lambda_2$ weighting. "row-select ($n = 1$)" and "column-select ($n = 1$)" are the naive first bounce operation also shown in the main paper. "single-channel" uses the single channel attention blocks in FLUX rather than the dual-channel blocks. "incoming ($n = 2$)" uses the incoming attention formulation of multi-bounce-attention (i.e. second bounce from a one-hot row vector). "incoming ($n = 3$)" is the same but with 3 bounces. "outgoing ($n = 3$)" uses a one-hot column vector with 3 bounces. "w/o $\lambda_2$" does not use the weighted $\lambda_2$ scheme. Notice the performance for outgoing attention degrades rapidly with more bounces ("outgoing ($n = 2$)*", "outgoing ($n = 3$)") compared to incoming attention ("incoming ($n = 2$)", "incoming ($n = 3$)"). This is because for outgoing attention, the steady-state eventually converges into emphasizing important "hubs", which usually tends to be background information. However, in early bounces ($n = 1, 2$) it out-performs the incoming attention, and we speculate this is because it captures better the target of this task: we are interested in finding out which image tokens "attend-to" the text token of interest (i.e. a many-to-one relationship).

We additionally report error bars in Table 8 that were computed as standard deviations over 5 runs with different seeds.

### C.2    Further Qualitative Results

See Figures 13 and 14 for more zero-shot segmentation results on ImageNet with comparisons to row- and column select and state-of-the-art results using ConceptAttention (Helbling et al., 2025). We argue that our raw segmentation masks are cleaner and more precise, which can be qualitatively measured by the threshold-agnostic mAP metric presented in the main paper. Additionally, in Figure 6 we show our multi-bounce attention can be used for segmenting different parts of an image by setting the initial one-hot vector to the corresponding text token. The text prompt used to create the image (FLUX-Schnell) was "cute black cat standing up wearing red boots and a bow tie, photorealistic, masterpiece, macro wildlife photography, dewy grass, closeup low angle, wildflowers, sun, shadows, depth of field, desaturated, film grain, low quality".

Table 8: **ImageNet segmentation.** Error bars for selected runs.

| Method | Acc | mIoU | mAP |
|---|---|---|---|
| Concept Attention | $81.06 \pm 0.01$ | $66.02 \pm 0.01$ | $88.43 \pm 0.00$ |
| Ours no $\lambda_2$ | $84.00 \pm 0.02$ | $69.99 \pm 0.05$ | $94.26 \pm 0.04$ |
| Ours | $84.11 \pm 0.03$ | $70.19 \pm 0.02$ | $94.32 \pm 0.03$ |

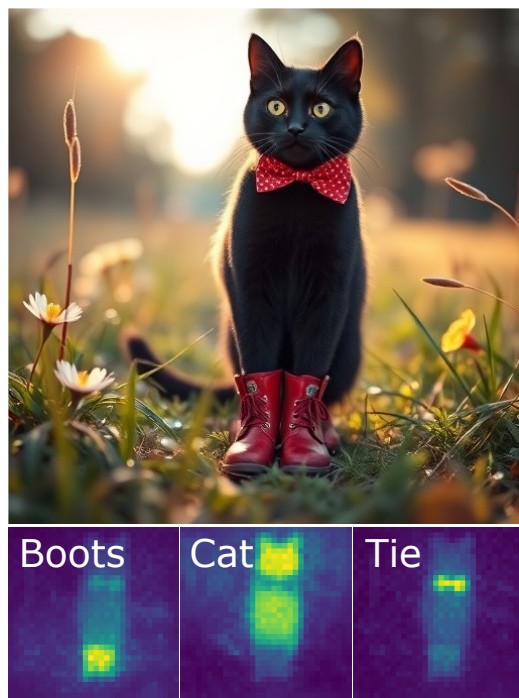

Figure 6: **Part segmentation.** We can segment different parts of the image by setting the initial one-hot vector to the corresponding text token.

## D Additional Results for Masking Most Influential Tokens

### D.1 Per-Model Results

In Figure 7 we show the results of performing the most-influential token masking experiment described in the main paper for individual models. The area-under-curve (AUC) metric in the main paper is computed over the curves.

### D.2 Per-Layer Results

In Figure 8, we show the result of choosing progressively more number of layers to mask for the most-influential token masking experiment conducted in the main paper. "6 (Ours)" indicates what we used in the paper. Layers are selected starting from the furthest from the output. The results are shown for DINOv2 (ViT-S/14) w/ registers over the same dataset used in the main paper. However, similar trends were observed for all architectures. We opted for masking tokens from only the first 50% of the layers to avoid distorting the output space and a too steep performance drop (which happens for $n > 6$), while still having an effect on the classification results for higher token mask percentage (which does not occur for $n < 6$).

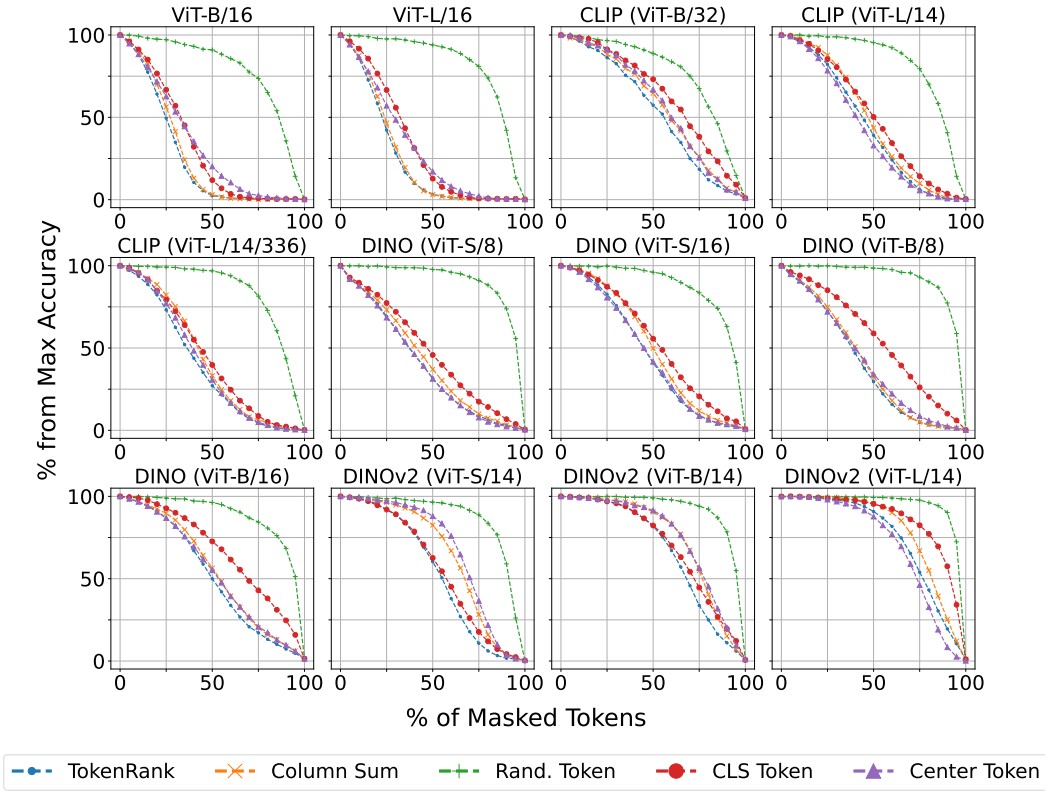

Figure 7: **Token masking.** Using TokenRank to mask out tokens better degrades performance on average across multiple architectures and training types.

Table 9: **Statistical test for $\lambda_2$ weighting.** The results of the Wilcoxon test show that $\lambda_2$ weighting is significantly better than uniform weighting.

| | Semantic Segmentation | | | Linear Probing | |
| --- | --- | --- | --- | --- | --- |
| Metric | Statistic | p-value | Model | Statistic | p-value |
| Acc | 15.0 | 0.031 | DINOv1 | 45.0 | 0.002 |
| mIoU | 15.0 | 0.031 | DINOv2 | 42.0 | 0.010 |
| mAP | 15.0 | 0.031 | CLIP | 45.0 | 0.002 |

# E    Additional Results for $\lambda_2$ Weighting

## E.1    Statistical Tests for $\lambda_2$ Weighting

In this section we provide more evidence that the magnitude of the second largest eigenvalue of the attention matrix correlates with the performance in both zero-shot segmentation (Section 5.1) and linear probing (Appendix B), as presented in the main paper. We illustrated this effect via weighted averaging of heads. To test the significance of our observation, we perform a paired data statistical test (Wilcoxon test) for both tasks. For this purpose, we define the null hypothesis: Uniform and $\lambda_2$ weighting have the same performance. And the one-sided alternative hypothesis: Weighting results in a better performance. Results of can be seen in Table 9. Rejecting the null hypothesis with $p < 0.05$ supports our statements.

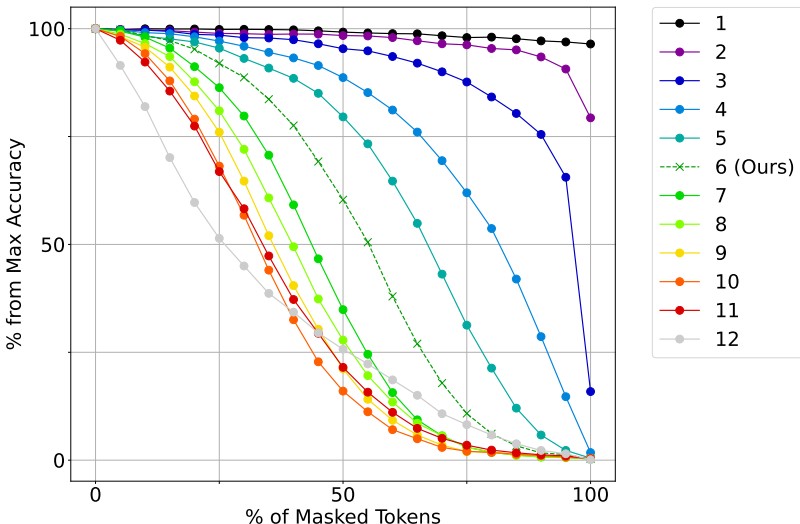

Figure 8: **Layer selection for DINOv2 (ViT-S/14).** Numbers indicate the number of attention layers being masked starting from the closest to the input. Masking fewer layers introduces noise into measurements, while masking more of them results in a faster drop in accuracy allowing less room for comparisons. We chose to mask 50% of the layers for all experiments; in this case "6 (Ours)" variation was used.

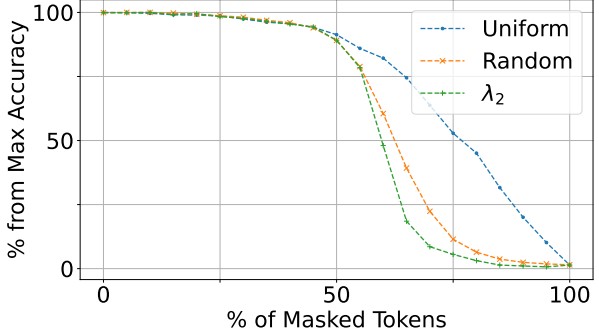

Figure 9: **Aggregating heads.** We used TokenRank to mask tokens while using different weighting schemes for the head dimension. $\lambda_2$ yields a larger drop in accuracy compared to uniform or random weighting.

### E.2 $\lambda_2$ Weighting for Masking Most Influential Tokens Experiment

We performed another token masking experiment similar to the one conducted in of the paper, only using TokenRank for determining the masking order. In this experiment, we compare using uniform weights per head (as in main paper), random head weights, and using $\lambda_2$ weighting. Results can be seen in Figure 9. Perhaps unsurprisingly, randomly weighting the heads performs better than uniform weights, as the transformer is more sensitive to individual heads being fully or almost fully masked out, rather than gradually masking out all heads together. Importantly, results indicate that $\lambda_2$ weighting can be used for increasing the degradation in accuracy, suggesting it can serve as a useful tool for determining importance of heads. This was also observed to improve zero-shot segmentation results (see Appendix C).

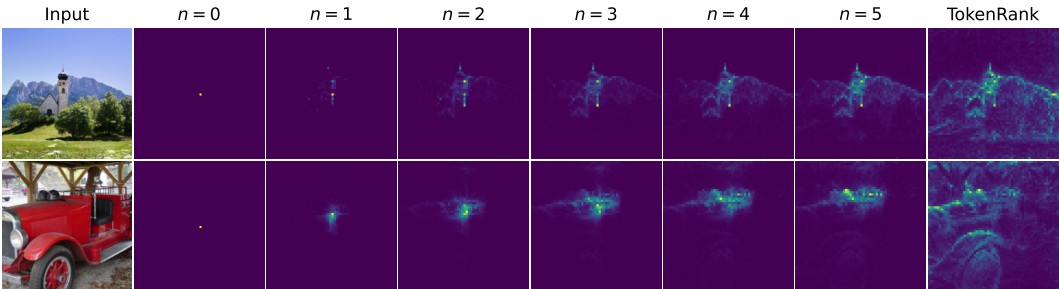

Figure 10: **Illustration of multi-bounce attention.** We visualize the first bounces and the steady state (*TokenRank*) when propagating through the Markov chain, defined by two attention matrices of two exemplary layers and heads of DINOv2. The initial state ($n = 0$) is defined by a one-hot vector.

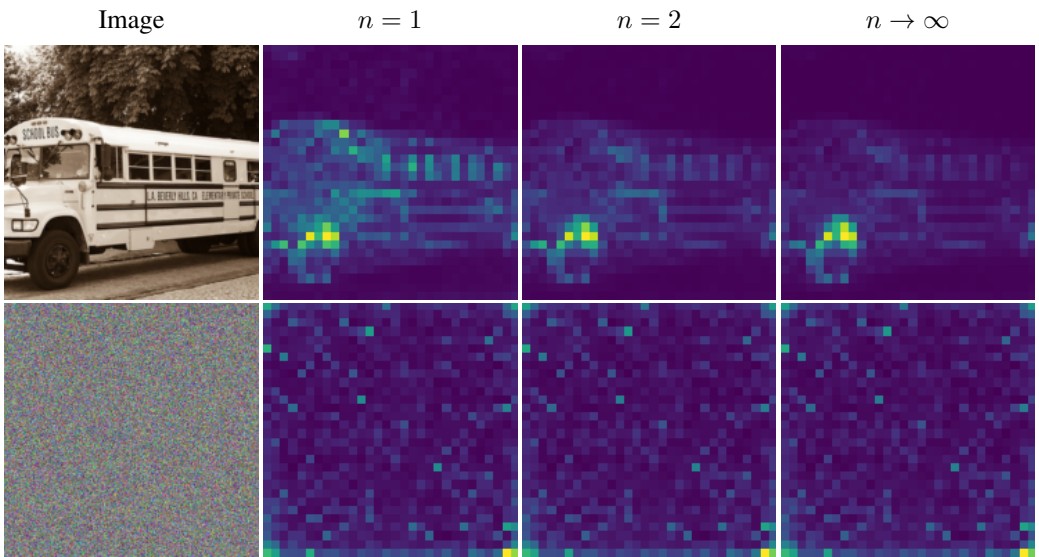

Figure 11: **Illustration of bouncing for image and random input.** Bouncing the attention signal sharpens the DINOv1 attention map for a realistic input image, supporting the existence of meta-stable states. On the contrary, a random input image results in a fast converging Markov chain, where intermediate bounces to not differ clearly from the first bounce and disperse rather than cluster. This observation is also reflected in the second eigenvalues: $\lambda_2 = 0.44$ for the real image and $\lambda_2 = 0.16$ for the random image, where a smaller second eigenvalue corresponds to a faster convergence.

## F    Illustration of Bouncing as Consolidation Mechanism

Figure 10 visualizes the first bounces and the steady state of two example images, heads and layers.

Figure 11 also illustrates that iterating over the Markov chain results in an iterative refinement of the map, consolidating meta-stable states. The steady state eventually visualizes the global incoming attention. This effect cannot be observed for an input image with random pixels (Figure 11, bottom). Instead, the Markov chain converges very quickly to a dispersed visualization after one step for a noisy input, as the attention matrix is not structured. We compute the visualizations by passing the images through DINOv1 and extracting the attention map in the 9th layer and 3rd head. Then, we compute the first bounce ($n = 1$) with a uniform initial vector, the 2nd bounce, and the steady state. The same effect can be seen in Figure 13, where the second bounce (*Ours*) is significantly cleaner and better adheres to the object boundaries compared to the first bounce (*Column Select*).

## G    More SAG Results

In Figure 12 we show more results comparing vanilla SD1.5 (Rombach et al., 2022), SAG (Hong et al., 2023) and SAG+TokenRank.

## H    Per-Head TokenRank Visualizations

We show more visualizations for TokenRank, column sum, Center Patch, and CLS token for various heads and layers in Figure 15 and Figure 16.

## I    Visualization Across Generation Timesteps

In Figure 17, we show generated image results of using FLUX-Dev with 50 timesteps. We computed the TokenRank (incoming attention) and average over every 5 consecutive timesteps shown in every column (first entries correspond to noisier timesteps, while last entries are more noise-free). The TokenRank visualizations remain stable through the timestep dimension. It can be observed how attention maps are getting sharper during the denoising process, illustrating that TokenRank can serve to visualize and analyze generative models.

## J    Used Compute

The image segmentation experiments with FLUX required around 1500 GPU hours, where one experiments takes around 50 GPU hours. Extracting all attention map visualizations for the linear probing experiment and training the linear classifier took around 300 GPU hours. We performed around 40 experiments for the masking experiment, where each experiment required around 30 GPU hours, resulting in around 1200 GPU hours in total. Finally, the SAG experiment took around 1000 GPU hours. In total, we estimate the use of around 4000 GPU hours for the whole study. For all computations, we used an internal GPU cluster consisting of NVIDIA A40, A100, H100, and RTX 8000 GPUs.

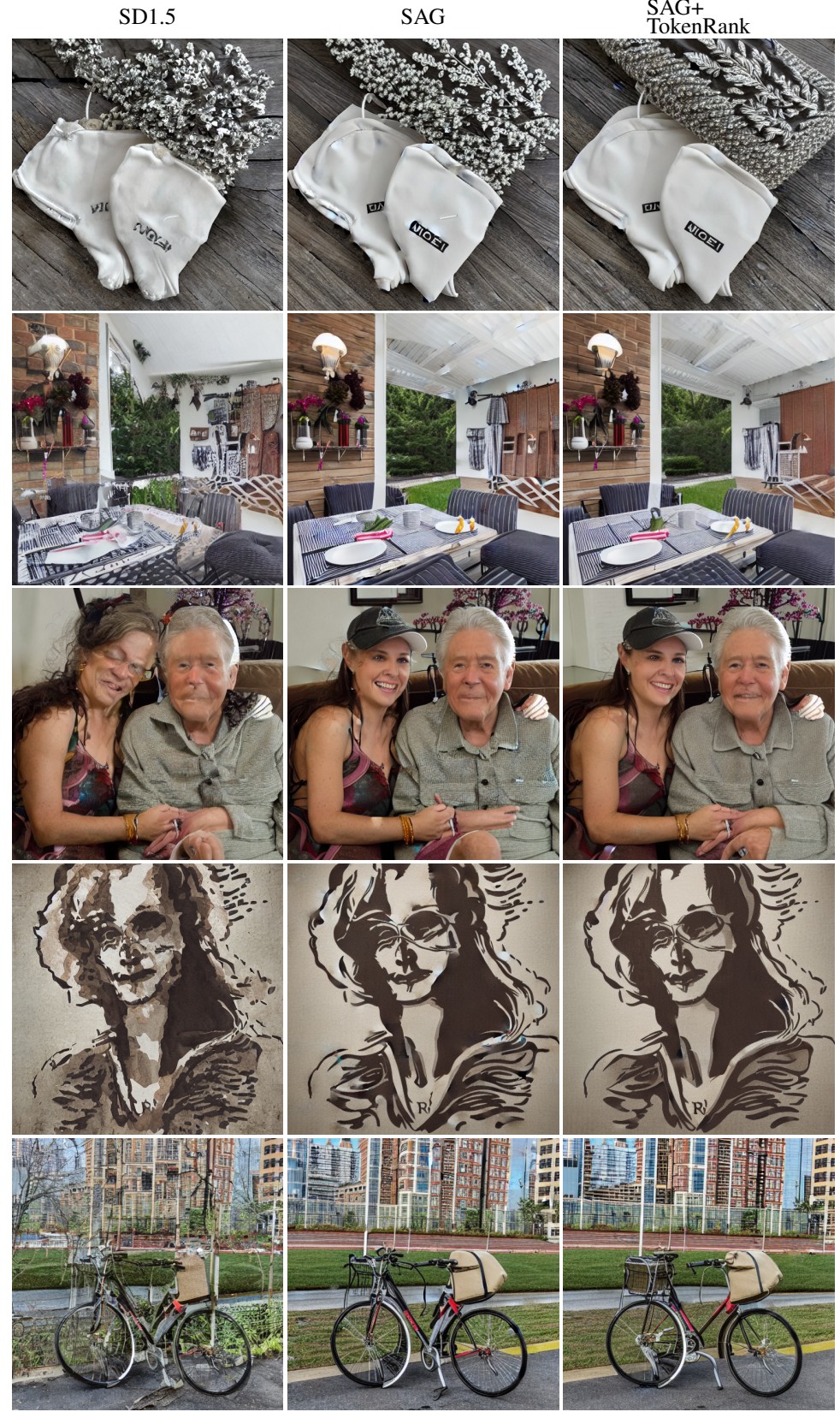

Figure 12: **Improvement of SAG.** Using TokenRank produces less artifacts and more structured images.

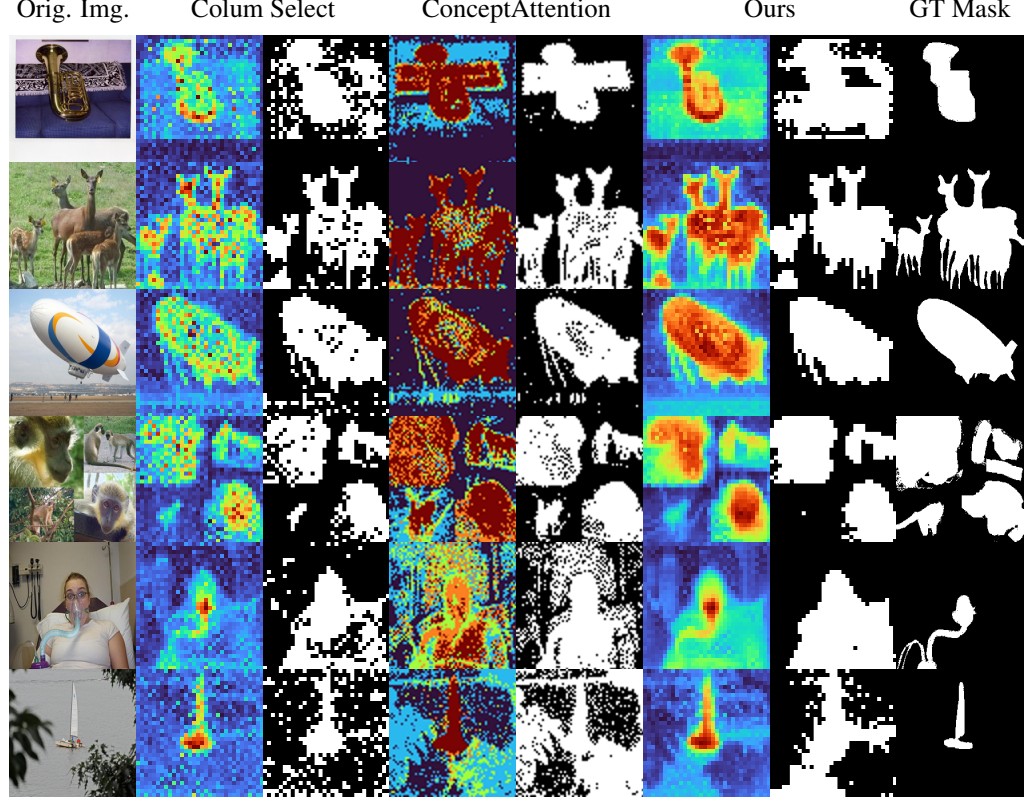

Figure 13: **ImageNet segmentation.** Our results are on par with state of the art ConceptAttention (Helbling et al., 2025).

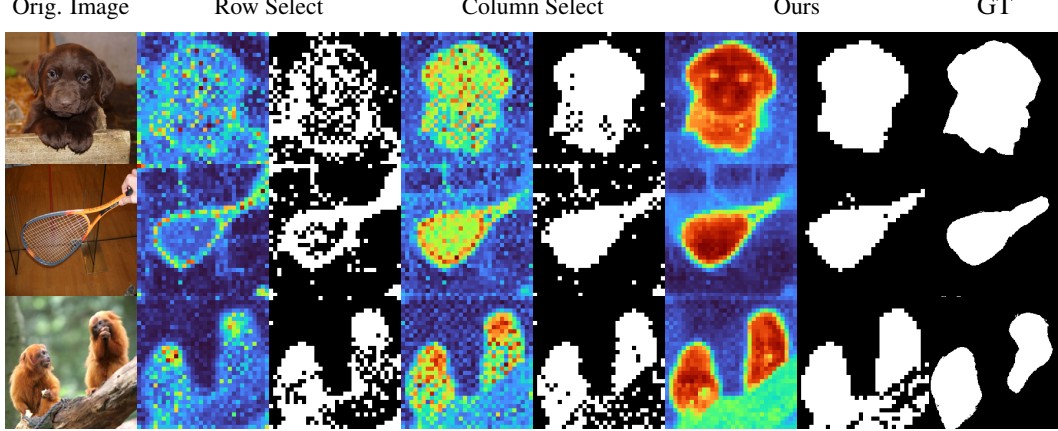

Figure 14: **ImageNet segmentation.** We visualize the raw attention output (colored) and the binary segmentation masks for the row- and column- select operations compared to utilizing multi-bounce attention.

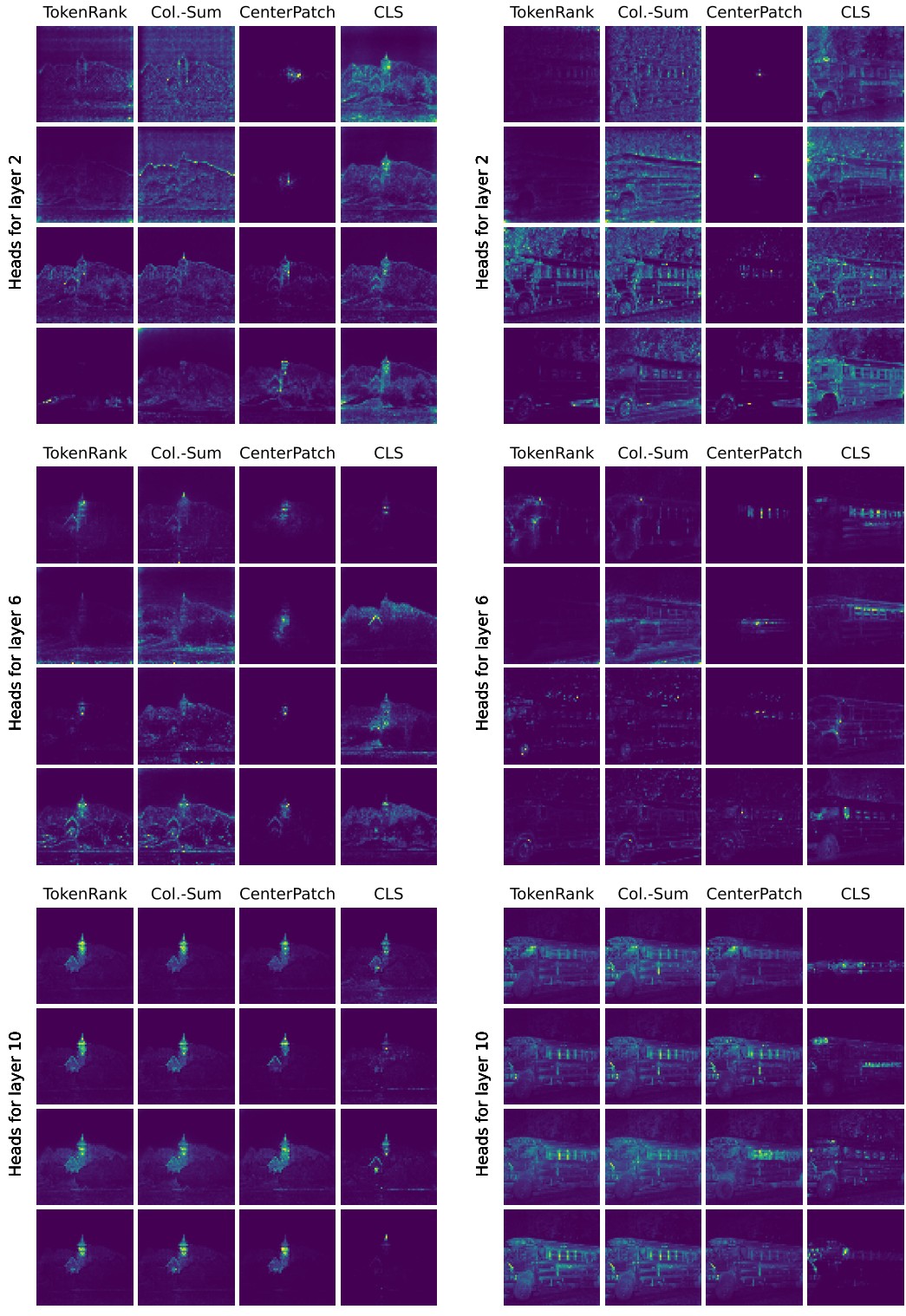

Figure 15: **Per-head visualizations of global incoming attention.** We plot with different visualization strategies for various layers and the first four heads for DINOv1 (ViT-B/8). Images correspond to the lower row of Figure 4 in the main paper.

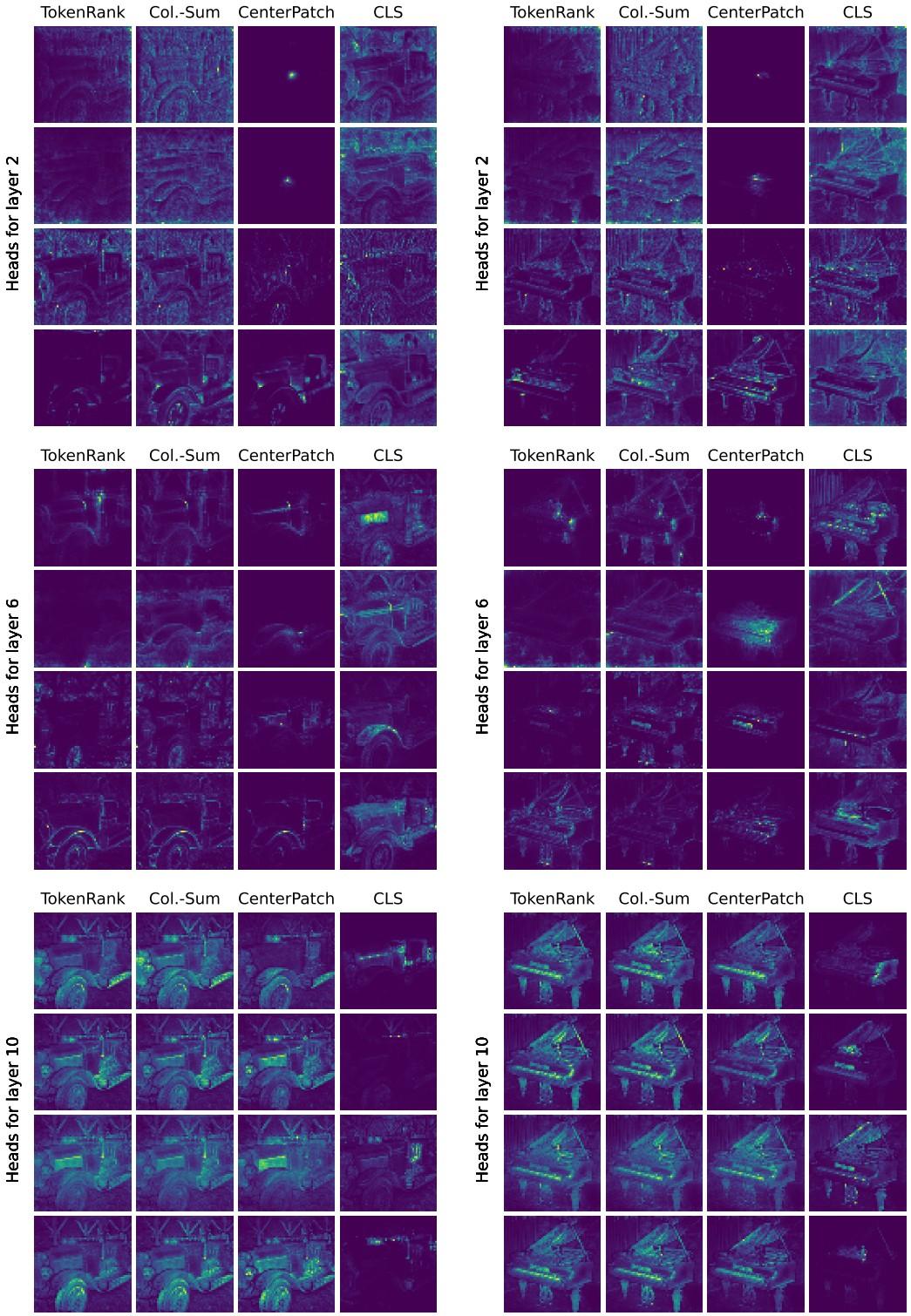

Figure 16: **Per-head visualizations of global incoming attention.** We plot with different visualization strategies for various layers and the first four heads for DINOv1 (ViT-B/8). Images correspond to the lower row of Figure 4 in the main paper.

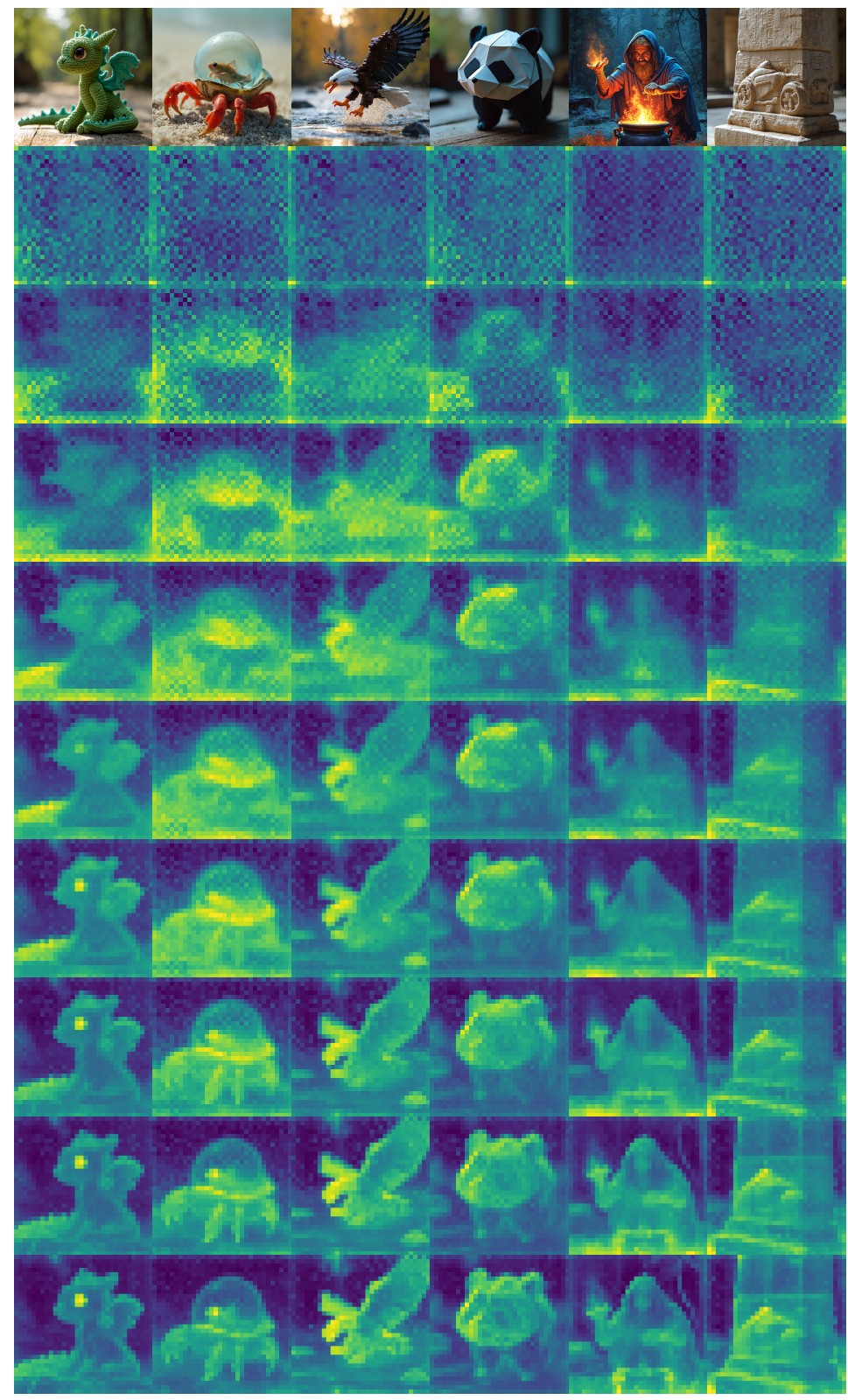

Figure 17: **Timestep stability.** Top: generated images using FLUX-Dev with 50 timesteps. Each column depicts the TokenRank (incoming attention) for increasingly denoised images during the diffusion process. Rows correspond to decreasing timesteps.

