# OpenReview forum: "Attention (as Discrete-Time Markov) Chains"
_NeurIPS.cc/2025/Conference — NeurIPS 2025 poster_

### Official Review · Reviewer_YchT · 2025-06-21

**Clarity:** 4
**Significance:** 3
**Originality:** 4
**Rating:** 5
**Confidence:** 3

**Summary:**

This work proposes to interpret the attention matrix as the transition matrix of a markov chain. The multi-hop attention can then be computed to obtain a token importance measure of higher order. It then proposes TokenRank----the steady state vector of the markov chain that offers a better global token importance measure and can be used to improve several downstream tasks.

**Questions:**

See weakness 1.

**Ethical Concerns:**

["NO or VERY MINOR ethics concerns only"]

**Final Justification:**

I think the method is novel and offers a new perspective to view the attention matrix that provides a good global representation. It would open up a lot potential future applications in different domains and potentially have a lot of impact.

**Limitations:**

See weakness 2 and weakness 3

**Paper Formatting Concerns:**

The citation format seems to be wrong.

**Quality:**

4

**Strengths And Weaknesses:**

Strength
1. This method is novel in a sense that no one has interpreted the attention matrix as a transition matrix before. It attempts to provide a framework to unify previous interpretations and further improve the quality of the representation, using well-studied knowledge in the markov chain.
2. The method is simple yet effective: By considering information from multiple hop of tokens, a better global representation can be built through simply multiplying the adjusted attention matrix iteratively.

Weakness
1. Efficiency: Although the operations seem simple, I would still like to know whether computing the TokenRank vector will have a cost in efficiency. I wonder how many iterations are needed on average to compute the steady state vector and how much time will they cost. When TokenRank is used in downstream tasks like zero-shot segmentation and unconditional image generation with SAG, how much time is added when using TokenRank?
2. Minor point not affecting judgement: The work only demonstrates application on vision transformer models. It would be nice if we can see the application of TokenRank on natural language processing tasks like sentiment analysis etc to see if there is any caveat or new discovery.
3. Exploration of TokenRank with multi-modal cross attention: I think this work itself is rather complete. Potential directions to explore would be seeing whether the representation generated by TokenRank can locate specific objects in local regions, instead of a global representation of the image presented in this work. Potential downstream tasks can be visual grounding or image and sentence matching with models like “Multi-Modality Cross Attention Network for Image and Sentence Matching”.

---

> ### Author Rebuttal · Authors · 2025-07-30
>
> We thank the reviewers for their constructive feedback and ideas. We are happy that the reviewers like the elegant and insightful framework (BdYW) that is grounded in both theoretical insight and empirical validation (pbyL), its effectiveness (BdYW,YchT) and novelty (BdYW,mHJ9,pbyL,YchT), and the clear and well-structured description (mHJ9,pbyL). The provided feedback helps us to improve the paper and we will address the individual concerns raised by reviewer YchT below.
>
> > Efficiency: Although the operations seem simple, I would still like to know whether computing the TokenRank vector will have a cost in efficiency. I wonder how many iterations are needed on average to compute the steady state vector and how much time will they cost. When TokenRank is used in downstream tasks like zero-shot segmentation and unconditional image generation with SAG, how much time is added when using TokenRank?
>
> TokenRank converges typically within 20 iterations for SD1.5. The $\lambda_2$ value can directly indicate how many iterations will be needed, though computing $\lambda_2$ currently takes more time than computing TokenRank. As for computation, we profiled SAG and it takes roughly 1.2x times to perform with TokenRank (averaged over thousands of samples). This relatively mild extra computational effort is because it involves a very efficient (vector,matrix) product operation, which is performed on the GPU. We implemented a batched version as well.
> For zero-shot segmentation, it takes 1.1x more times to perform 2 bounces compared with the baseline with 1 bounce (e.g. column- select.
>
> >It would be nice if we can see the application of TokenRank on natural language processing
>
> Thanks for this comment. Similarly to our response to reviewer BdYW, while we appreciate the impact of our study might have been stronger by validating and experimentation with other modalities, we intentionally focused on extensive and rigorous experimentation on one specific domain for the purpose of anchoring this interpretation as useful and impactful. We consider exploring the usefulness of our framework for other domains such as NLP, motion or video would be an exciting future work prospect.
>
> > Exploration of TokenRank with multi-modal cross attention: I think this work itself is rather complete. Potential directions to explore would be seeing whether the representation generated by TokenRank can locate specific objects in local regions, instead of a global representation of the image presented in this work.
>
> This is a great direction. While TokenRank is by nature a global indicator, we presented multi-bounce as its “counter-part” that allows for local probing. One interesting way of fully exploiting the Markov chain formulation and “localizing” TokenRank, is to introduce the famous “personalization” vector from Google’s PageRank paper. This vector was originally designed to allow ranking pages of the internet but take into account user preferences (e.g. if a user like sports -> rank sports related pages higher). In our case, such a vector could potentially be used to “personalize” TokenRank, such that specific semantic tokens / groups of tokens are ranked higher (e.g. “cat” related tokens). We have not validated this approach, but believe it could be used to solve more elaborate and localized downstream tasks.

---

> > ### Comment · Reviewer_YchT · 2025-08-01
> >
> > We thank the author for providing additional efficiency numbers on their methods. We would keep the rating as it is.

---

### Official Review · Reviewer_pbyL · 2025-07-03

**Clarity:** 3
**Significance:** 4
**Originality:** 4
**Rating:** 5
**Confidence:** 5

**Summary:**

This paper proposes a novel interpretation of attention matrices in transformers as discrete-time Markov chains (DTMCs), allowing for principled analysis of attention dynamics. By treating the attention matrix as a transition matrix, the authors are able to introduce concepts such as multi-bounce attention to capture higher-order token interactions, and TokenRank, a PageRank-inspired steady-state vector measuring global token importance. The framework leads to improved zero-shot segmentation performance, sharper visualizations of attention, and enhancements in generative tasks like diffusion-based image synthesis.

**Questions:**

The authors need to improve the readability of the work. My central concern is just how many results the authors are trying to show, but they end up falling short of satisfaction in each section.

1. The work currently lacks a lot of details in the experiments section. Please pull all of this out of supplementary purgatory and put it into the main paper.
2. No error bars are provided, therefore no claims about the significance of the results can be made. The authors claim in the checklist that experiments with error bars are **not** provided, but then justify this by saying that "we run the segmentation results several times..." So the authors have a way to provide error bars but choose not to? I don't agree with the "computational expense" excuse to not provide error bars. At least, discuss this in the main paper, not the checklist!

If the authors can address these two claims in their rebuttal (and therefore an edit of their paper), that will be satisfactory.

**Ethical Concerns:**

["NO or VERY MINOR ethics concerns only"]

**Final Justification:**

The authors provided error bars and statistical tests for their results. This was my biggest gripe, and they addressed it!

Further, the authors were able to provide clarifications about their wording, and they have promised to trim the extraneous results and move the salient ones from the supplement to the main body.

**Limitations:**

Yes

**Quality:**

3

**Strengths And Weaknesses:**

I like this paper and found parts of it to be intuitive. However, there are areas in which I struggled to get sufficient detail. See below!

# Strengths
1.  The paper is clear, well-structured, and grounded in both theoretical insight and empirical validation.
2. The interpretation of attention as a DTMC unifies many existing operations (row select, column sum, averaging) under a cohesive framework (L1-14, L36-43) and cleanly ties into the well-understood theory of PageRank.
3. The proposed techniques yield state-of-the-art performance in zero-shot segmentation (Table 1), improved visualization fidelity (Figure 4), and boosts in generative tasks like SAG (Table 3).

# Weaknesses
1. L7-8: "set of metastable states where the attention clusters while noisy attention scores tend to disperse." This phrasing is awkward and could benefit from clarification.
2. L12-13: "We demonstrate that using it brings improvements in unconditional image generation." It would be more informative to include specific quantitative results (e.g., improvement in FID/IS scores).
3. L37: "(non-negative)". I'm not sure why this is in parentheses!
4. L55/223: "2nd" should be spelled out
5. L42-43: "we argue that considering direct token relationships is not necessarily best for evaluating importance." This claim would benefit from a few more words. What characteristic should be considered instead/what is your method highlighting that these others are not?
6. Section 3.1 discusses convergence to $v_{ss}$​, but it would be nice to tie this into the formalized equations where $v_{ss}$​ is explicitly shown as a left-hand side variable.
7. L225-227: "we hypothesize that larger values are statistically correlated with the amount of important information..." No statistical test is presented to evaluate this hypothesis. Rephrase this as a nullable hypothesis, and then show the statistical significance of your findings to see if this hypothesis holds or does not hold.
8. L255 + others: "See supplementary for an illustrative example." So many critical details are offloaded to the supplement. The supplement is not even part of the main PDF, which drastically reduces the ease at which I can understand what's going on in the paper. Further, please reference specific sections in the supplement rather than vaguely referencing "the supplementary".
9. L252-253: "indicating that our method is on par and arguably better than the previous state-of-the-art..." All of the comparisons in this paper lack error bars. Going back to weakness 7, how can anything be "arguably" better? It either is or isn't, and you can show that with error bars and a statistical test. These experiments results seem compelling but need significance to underline the claims.
10. In general, the authors attempt to show too many use cases, which dilutes the focus. Section 5.3, which was highlighted in the abstract (!) gets minimal exposition. I would consider trimming less central experiments to allow for deeper analysis of key results. Right now the paper reads pretty thin.
11. This work is not the first to relate attention and Markov Chains. It's worth citing Ildiz et. al [0] and discussing in the related works where this work differs.
12. I'm not sure if I agree with an answer of "NA" to question 3 on the checklist. The authors are providing new theoretical results (tying attention to DTMC), but because they've left themselves no space, they are not able to clearly present this theory in the paper.
13. Checklist question 4 is marked "yes", but even in the supplement, there is not enough detail about how the authors ran their experiments to redo them. Which experiments ran on which GPU (of the 4 types used)?


[0] Ildiz, M. E., Huang, Y., Li, Y., Rawat, A. S., & Oymak, S. (2024). From self-attention to markov models: Unveiling the dynamics of generative transformers. arXiv preprint arXiv:2402.13512.

---

> ### Author Rebuttal · Authors · 2025-07-30
>
> We thank the reviewers for their constructive feedback and ideas. We are happy that the reviewers like the elegant and insightful framework (BdYW) that is grounded in both theoretical insight and empirical validation (pbyL), its effectiveness (BdYW,YchT) and novelty (BdYW,mHJ9,pbyL,YchT), and the clear and well-structured description (mHJ9,pbyL). The provided feedback helps us to improve the paper and we will address the individual concerns raised by reviewer pbyL below.
>
>
> > L7-8: "set of metastable states where the attention clusters while noisy attention scores tend to disperse." This phrasing is awkward and could benefit from clarification.
>
> This will be fixed to spell: “Our key observation is that tokens linked to semantically similar regions form \textit{metastable} states, i.e., regions where attention tends to concentrate, while noisy attention scores dissipate.”
>
> > L12-13: "We demonstrate that using it brings improvements in unconditional image generation." It would be more informative to include specific quantitative results.
>
> We agree it is useful to mention in what aspects did the unconditional generation improve, but since this application is only one of many downstream tasks that our conceptual framework improves, we will fix this sentence to spell:
> “We show that TokenRank enhances unconditional image generation, improving both quality (IS) and diversity (FID)”.
>
>
> > L37: "(non-negative)". I'm not sure why this is in parentheses!
>
> We will remove the parentheses.
>
> > L55/223: "2nd" should be spelled out
>
> Will be fixed.
>
> > L42-43: "we argue that considering direct token relationships is not necessarily best for evaluating importance." This claim would benefit from a few more words.
>
> Thanks. This paragraph draws an analogy between PageRank and our framework, where for webpages, simply counting incoming links will yield a bad metric of overall importance. Similarly, simply using direct attention (row- or column-select) to measure global relevance of a token underperforms compared to propagating it through the chain, as we show in various experiments. This considers indirect attention propagated through multiple bounces, eventually taking into account all indirect attention effects (steady-state). We will refine the paragraph to reflect this.
>
> > Section 3.1 discusses convergence to $v_{ss}$, but it would be nice to tie this into the formalized equations where $v_{ss}$ is explicitly shown as a left-hand side variable.
>
> While we can formulate $v_{ss}$ in closed form by taking the limit $v_{ss} = \lim_{n\rightarrow\infty}([1 .. 0 .. 0]\cdot A^n)$, this would be slightly less elegant since we wish for the readers to establish an intuition and familiarity with the power-method approach for later on. We believe it is quite straight forward to see why $v_{ss}$ is reached using the power method when $n\rightarrow\infty$.
>
> > L225-227: "we hypothesize that larger values are statistically correlated with the amount of important information..." No statistical test is presented to evaluate this hypothesis.
>
>
> We appreciate this comment and agree with the suggestion.
> We hypothesize that the magnitude of the second largest eigenvalue of the attention matrix correlates with the downstream task performance (zero-shot segmentation and linear probing). We illustrated this effect via weighted averaging of heads.
> To test the significance of our observation, we perform a paired data statistical test, ie, the Wilcoxon test for semantic segmentation and linear probing. For this purpose, we define the null hypothesis: Uniform and $\lambda_2$ weighting have the same performance. And the one-sided alternative hypothesis: $\lambda_2$ weighting results in a better performance.
>
>
> **Semantic segmentation**
>
> | Metric               | statistic | p-value |
> |---------------------|------------|---------|
> | Acc  | 15.0       | 0.031   |
> | mIoU  | 15.0       | 0.031   |
> | mAP  | 15.0       | 0.031   |
>
>
> **Linear probing**
>
> | Model               | statistic | p-value |
> |---------------------|------------|---------|
> | DINOv1  | 45.0       | 0.002   |
> | DINOv2  | 42.0       | 0.010   |
> | CLIP    | 45.0       | 0.002   |
>
> Rejecting the null hypothesis with $p<0.05$ supports our statements.
>
> >L255 + others: So many critical details are offloaded to the supplement.
>
> We apologize for the inconvenience, and will use exact references to the supplementary.
>
> > L252-253: "indicating that our method is on par and arguably better than the previous state-of-the-art..." All of the comparisons in this paper lack error bars. Going back to weakness 7, how can anything be "arguably" better?
>
> Thank you for pointing this out. We apologize for the unclear wording. The usage of "arguably" was not related to statistical significance. Instead, while Acc and mAP are largely better, we underperform ConceptAttention for mIOU. We argue that we are better than ConceptAttention (despite the drop in mIOU), as the threshold-agnostic metric mAP shows a large gap in performance.
> The qualitative results in the supplementary (Fig.7) also illustrate how our attributions are smooth and intuitive, while the ones by ConceptAttention appear to be noisy and over-confident.
>
> However, following the comment about missing error bars, we computed mean and std for the following results:
>
> **Semantic segmentation**
>
>
> | Model                              | Acc             | mIoU             | mAP             |
> |------------------------------------|------------------|------------------|-----------------|
> | ConceptAttention                   | $81.06 \pm 0.01$ | $66.02 \pm 0.01$ | $88.43 \pm 0.00$ |
> | Ours without $\lambda_2$ weighting | $84.00 \pm 0.02$     | $69.99 \pm 0.05$    | $94.26 \pm 0.04$  |
> | Ours | $84.11 \pm 0.03$     | $70.19 \pm 0.02$    | $94.32 \pm 0.03$  |
>
>
> The error bars (std) are computed over 5 runs with different seeds. We reproduced ConceptAttention using their implementation. The results are lower than the reported ones in their paper. However, the experiments show that different seeds do not have a significant influence on the computed results.
>
>
> **Linear probing** (Tab. 2)
>
> | Mode        | DINOv1   | DINOv2   | CLIP     |
> |-------------|---------------------|----------------------|---------------------|
> | Randompatch | $67.22 \pm 3.12$    | $63.32 \pm 6.04$     | $57.65 \pm 2.66$    |
> | Centerpatch | $66.32 \pm 3.08$    | $75.34 \pm 1.85$     | $68.42 \pm 2.66$    |
> | Colsum      | $75.64 \pm 2.74$    | $90.76 \pm 2.21$     | $73.73 \pm 2.98$    |
> | TokenRank        | $77.31 \pm 2.50$    | $92.73 \pm 1.51$     | $73.88 \pm 2.86$    |
> | CLS         | $81.53 \pm 2.44$    | $94.07 \pm 0.97$     | $72.46 \pm 3.40$    |
>
> Furthermore, we performed a Wilcoxon test with the null hypothesis that the competing column sum operation results in the same accuracy as TokenRank and the one-sided alternative hypothesis that the accuracy is lower. We reject the null hypothesis with p=0.03 (DINOv1) and p=0.03 (DINOv2).
>
> We believe the **masking results** (Sec. 5.4) are already stable since they were computed based on a set of models.
>
>
> >I would consider trimming less central experiments to allow for deeper analysis of key results.
>
> Thank you for this suggestion.
> We concur and we will move the probing experiments to the supplementary materials to allow a more elaborate discussion of the key results, particularly considering the effectivness of consdering additional bounces (semantic segmentation) and the evaluation of the global importance via TokenRank.
>
>
> >It's worth citing Ildiz et. al [0] and discussing in the related works where this work differs.
>
> Thank you. Ildiz et. al [0] has slipped under our radar and considers modelling a 1-layer transformer as a Markov chain, and come to the significant conclusion that the *output* space is generated by transitioning through a recoverable matrix $A$ learned by the transformer, weighted by the frequency of tokens in the *input* prompt. This means that positional encoding for this case is not important, as the transformer merely assigns probability vectors for every token in the embedding and multiplies them by the frequency of the input prompt to generate a result. There are two main fundamental differences between our studies. First, we focus on large pretrained transformers and model each individual self-attention matrix as a Markov chain, while Ildiz et. al focus on a degenerate case of a single layer, and model the entire transformer pipeline as a Markov chain. While this allows Illdiz et. al to draw some conclusions and explanations regarding the behavior of larger transformers, we explicitly show how our framework allows for the interpretation of existing, pretrained large transformers and for a better performance in various tasks. Secondly, we leverage more of the machinery of the Markov chain theory, including multiple transitions and steady state analysis. This was not considered previously.
>
> We will add this explanation to the related work section.
>
> >The authors are providing new theoretical results
>
> We do not believe our study provides any new theoretical results. Our mathematical model and its application are well-understood and explored in other fields. Casting attention into these frameworks constitutes novelty, but not in terms of theory.
>
> >there is not enough detail about how the authors ran their experiments to redo them. Which experiments ran on which GPU (of the 4 types used)?
>
> We politely disagree. We do believe the supplement provides details of how to reproduce our experiments, and do not think the GPU type has any effect on our results as long as the models fit into memory (other than the random seed, and performance speed). Furthermore, we are committed to publishing the full implementation of our framework, which will allow accurate reproduction of results. Please note we will also move substantial parts of the implementation details into the main paper, as suggested.

---

> > ### Comment · Reviewer_pbyL · 2025-08-02
> >
> > Thank you for your detailed rebuttal. I'm happy with what you provided, and I hope that you include your newly derived error bars and statistical tests in the paper, as I do believe they make it a lot stronger. I will revise my score.
> >
> > I will, however, politely grumble about your polite disagreement. Even if you think the GPU type does not have any effect on the results (and you are probably correct), what is the harm in providing the details? In the off-chance you are wrong, you will save some poor graduate student potentially hours of frustration trying to figure out what's going on. You will also be more in-line with the spirit of the NeurIPS checklist. I would urge you to please include the details if you have them as the effort is low but the contribution to the community is high.

---

> > > ### Author Response · Authors · 2025-08-04
> > > **Effect of GPU hardware**
> > >
> > > The experiments were performed using A40 and A100 GPUs. Due to the cluster environment, we, unfortunately, cannot trace back which experiment was run on what exact GPU.
> > > However, the various runs with different seeds for the error bar computation were performed on different GPUs.
> > > The low standard deviation supports that using different GPUs does not result in a significant variation of the results.

---

### Official Review · Reviewer_mHJ9 · 2025-07-03

**Clarity:** 4
**Significance:** 3
**Originality:** 3
**Rating:** 4
**Confidence:** 3

**Summary:**

In this work, the authors present a new interpretation for the attention matrix by modeling it as a discrete-time Markov chain, providing fresh insights into the mechanism of attention. Specifically, this paper treats the attention matrix as a state transfer matrix and consider the stable state of it. In this way, the proposed method brings improvements in unconditional image generation and zero-shot image segmentation.

**Questions:**

See Weaknesses. If the authors are able to address all or some of the weaknesses, I would consider raising my score.

**Ethical Concerns:**

["NO or VERY MINOR ethics concerns only"]

**Final Justification:**

My main concerns have been addressed. The authors compare their method with [1] and show further improvements. So I raise my score from 3 to 4.

**Limitations:**

See Weaknesses.

**Quality:**

3

**Strengths And Weaknesses:**

Strengths：

1. This paper is easy to follow, with a clear description of the proposed method and the problem it aims to address. The solution presented by the authors is also reasonable.
2. The results presented by the authors show that their method has clear advantages over other approaches.
3. To my knowledge, this is the first work to consider about the stable state of attention matrix.

Weaknesses：
1. In Section 4.1, the authors describe existing interpretations of attention mechanisms. However, I find this description to be incomplete, as it does not mention [1], which merges attention maps using KL divergence. Moreover, the comparative experiments do not include a comparison with this method. If the authors believe that this method is not relevant to the present work, they may provide a rebuttal.

2. The experimental results shown in Figure 6 of the supplementary material are convincing. However, I hope the authors can provide some visualizations of the feature maps or attention maps to further enhance the credibility of their results.

3. In Figure 3, the authors only present some basic attention visualization methods, while most of the quantitative comparisons are provided in the supplementary material. Although this does not affect the review process, if the paper is accepted for publication, it would be necessary to include more qualitative results directly in Figure 3. Moreover, the comparative methods included in the supplementary material appear to be insufficient.

4. In Figure 5, the authors demonstrate the effectiveness of the SAD method on the SD1.5 architecture. However, since SD1.5 is based on a UNet architecture, it would be valuable to include qualitative results on methods utilizing DiT-based architectures, such as PixArt or Flux.

[1] Diffuse, Attend, and Segment: Unsupervised Zero‑Shot Segmentation using Stable Diffusion

---

> ### Author Rebuttal · Authors · 2025-07-30
>
> We thank the reviewers for their constructive feedback and ideas. We are happy that the reviewers like the elegant and insightful framework (BdYW) that is grounded in both theoretical insight and empirical validation (pbyL), its effectiveness (BdYW,YchT) and novelty (BdYW,mHJ9,pbyL,YchT), and the clear and well-structured description (mHJ9,pbyL). The provided feedback helps us to improve the paper and we will address the individual concerns raised by reviewer mHJ9 below.
>
> > In Section 4.1, the authors describe existing interpretations of attention mechanisms. However, I find this description to be incomplete, as it does not mention [1], which merges attention maps using KL divergence. Moreover, the comparative experiments do not include a comparison with this method.
>
> We thank the reviewer for this reference. While this method does not provide any new interpretation of the attention maps per se (KL-divergence is used to measure similarity between upsampled attention maps), we agree it solves the same downstream task with partially similar toolkits (self-attention matrices, treating rows as probability distributions, aggregation of heads, etc.). We therefore evaluate [1] on our benchmark and found that our work outperforms [1] significantly:
>
> | Method | mAcc   | mIoU   | mAP    |
> |--------|--------|--------|--------|
> | [1]    | 65.41  | 52.12  | N/A    |
> | Ours   | 84.12  | 70.20  | 94.29  |
>
> For [1], mAP cannot be computed since the method does not provide raw scores.
>
> Note: we acquired class-specific semantic segmentation labels from the predicted labels using the Hungarian algorithm, following the original author's implementation.
>
> Additionally, inspired by the reviewer’s comment and in line with our goal of demonstrating the versatility of our framework, we incorporated TokenRank into the framework of [1]. Specifically, instead of sampling anchors from a uniform grid as in the original study the authors used a uniform grid to initialize the anchors used as seeds for their proposal algorithm. Instead, we sample anchors according to their TokenRank importance, with suppression to avoid repeatedly selecting the same location. This “smart” grid strategy leads to substantial improvements over the original approach, as demonstrated on COCO-Stuff-27 benchmark from [1]:
>
> | Method                     | mAcc   | mIoU   |
> |----------------------------|--------|--------|
> | Uniform Grid               | 72.50  | 43.60  |
> | "Smart" Grid (TokenRank)   | 84.97  | 44.87  |
>
>
> This new experiment further strengthens our study by demonstrating the versatility of our framework. This experiment also shows our framework is orthogonal to other solutions leveraging self-attention for solving downstream tasks. We once again would like to thank the reviewer for introducing us to this study.
>
> We will include the results in the updated manuscript.
>
>
> > The experimental results shown in Figure 6 of the supplementary material are convincing. However, I hope the authors can provide some visualizations of the feature maps or attention maps to further enhance the credibility of their results.
>
> Thank you for the suggestion. While we did include additional visualizations in the supplementary (Fig. 7 for semantic segmentation and Fig. 8 for TokenRank), we will add further examples including more visualizations of the bouncing operation across different initial states, models, images, layers, and heads, as well as additional semantic segmentation examples.
>
> >it would be necessary to include more qualitative results directly in Figure 3. Moreover, the comparative methods included in the supplementary material appear to be insufficient.
>
> We will add more qualitative results including comparisons to recent methods in Figure 3 and more results in the supplementary.
>
> > In Figure 5, the authors demonstrate the effectiveness of the SAD method on the SD1.5 architecture. However, since SD1.5 is based on a UNet architecture, it would be valuable to include qualitative results on methods utilizing DiT-based architectures, such as PixArt or Flux.
>
> We agree that applying our approach to DiT-based architectures would strengthen the work. However, SAG was specifically designed and thoroughly tested for Stable Diffusion and is not readily compatible with models like PixArt, and certainly not FLUX (due to the lack of an unconditional branch). Our SAG experiment serves as a representative use case, demonstrating that TokenRank captures global importance more effectively than the column sum operation in a generative downstream task. We will mention the other models in the discussion.

---

> > ### Comment · Reviewer_mHJ9 · 2025-08-05
> > **Raise my score from 3 to 4**
> >
> > My main concerns have been addressed. The authors compare their method with [1] and show further improvements. So I raise my score from 3 to 4.

---

### Official Review · Reviewer_BdYW · 2025-07-05

**Clarity:** 3
**Significance:** 3
**Originality:** 3
**Rating:** 4
**Confidence:** 2

**Summary:**

This paper presents a novel and theoretically grounded interpretation of attention matrices as discrete-time Markov chains (DTMCs), introducing operations like multi-bounce attention, TokenRank, and \lambda_2-weighted head averaging. The approach demonstrates state-of-the-art performance in zero-shot segmentation and improves unconditional image generation. The work is well-motivated, methodologically sound, and empirically validated.

**Questions:**

See weaknesses.

**Ethical Concerns:**

["NO or VERY MINOR ethics concerns only"]

**Limitations:**

More limitations should be discussed.

**Quality:**

3

**Strengths And Weaknesses:**

Strengths:

1.	Framing attention matrices as DTMCs is elegant and insightful. This allows modeling indirect attention paths (via Markov transitions), moving beyond prior work limited to direct interactions.

2.	Extensive experiments are conducted to show its SOTA performance on zero-shot segmentation and the effectiveness of TokenRank.

Weaknesses:

1. Experiments are limited to image tasks (image segmentation/generation). More validation in NLP or multimodal settings will be helpful to illustrate its effectiveness and domain generality.

2.	More ablation studies to evaluate the contributions of individual components (e.g., \lambda_2 vs. multi-bounce) are suggested.


3.	The discussions about the limitations are insufficient.

---

> ### Author Rebuttal · Authors · 2025-07-30
>
> We thank the reviewers for their constructive feedback and ideas. We are happy that the reviewers like the elegant and insightful framework (BdYW) that is grounded in both theoretical insight and empirical validation (pbyL), its effectiveness (BdYW,YchT) and novelty (BdYW,mHJ9,pbyL,YchT), and the clear and well-structured description (mHJ9,pbyL). The provided feedback helps us to improve the paper and we will address the individual concerns raised by reviewer BdYW below.
>
> >More validation in NLP or multimodal settings will be helpful to illustrate its effectiveness and domain generality.
>
> Thank you for this comment. While we fully appreciate our study's impact would have been stronger by validating with other modalities, we intentionally focused on extensive and rigorous experimentation on one specific domain for the purpose of anchoring this interpretation as useful and impactful. Exploring the usefulness of our framework for other domains such as NLP, human motion, or video would be an exciting future work prospect.
>
> > More ablation studies to evaluate the contributions of individual components are suggested.
>
> We will move the ablation table (Tab. 1 in the supplementary, for semantic segmentation) to the main paper to make it easier for readers to assess the contribution of individual components. If there are specific ablations the reviewer is interested in that do not appear in the table, we commit to providing them.
>
>
> > The discussions about the limitations are insufficient.
>
> We agree with the suggestion to discuss limitations in further detail. The main drawbacks of using our framework are:
> - Our formulation is limited to square matrices (e.g. self-attention / hybrid-attention blocks). Therefore, cross-attention blocks do not naturally fit into our Markov chain formulation due to the existence of non-accessible states. There could potentially be a way to resolve this by introducing dummy states with uniformly distributed transition probability.
> - The propagation of attention requires to process the attention maps post-process in an “offline” manner. We did not explore (and tend to be skeptical) that our framework can be leveraged to replace or improve a pre-trained transformer to perform its task better.
> - Computing the 2nd eigenvalues $\lambda_2$ is very slow due to the inability to leverage rapid decomposition methods that rely on symmetry or PSD. Computing multiple bounces or the TokenRank on the other hand is quite performant, as it relies on vector matrix product operations, and convereges typically after 10 to 20 iterations.
> - We observe that the performance gains with TokenRank are smaller for transformers where the attention map is less structured. Consider, e.g., the masking experiment in Tab.4, where TokenRank brings a signficantly larger gain for DINOv2 than for a supervised ViT.
>
> These limitations will be incorporated into the revised manuscript.

---

> > ### Comment · Reviewer_BdYW · 2025-08-07
> >
> > I am grateful to the authors for their response. The rebuttal addresses my concerns, and I will retain my initial evaluation.

---

### Note · Authors · 2025-08-11

Dear AC and Reviewers,

we would like to reiterate our appreciation for the quality of the reviews and comments that were made, and reaffirm our commitment to open-sourcing our implementation upon acceptance.
We were delighted to see all reviewers were satisfied with the rebuttal and either maintained or increased their scores.

The review process has strengthened our submission by conducting more experiments to investigate the statistical significance of our results when comparing to previous works and ablated components. Statistical tests confirmed that the paper's claims are stable.

Furthermore, motivated by comparisons with [1], we demonstrated that TokenRank can be employed to refine their grid sampling strategy, thereby providing additional evidence that our framework enhances downstream task performance through a lightweight post-processing procedure.

Lastly, a more detailed discussion including limitations will be presented, to support readers in better understanding the scenarios in which our framework is applicable.

All new results and discussed amendments will appear in the revised manuscript.


[1] Diffuse, Attend, and Segment: Unsupervised Zero‑Shot Segmentation using Stable Diffusion

---

### Decision · Program_Chairs · 2025-09-17

**Decision:**

Accept (poster)

**Comment:**

This paper proposes a discrete-time markov chain of attention matrix in transformers. The common operations in transformers are re-formulated from the markov chain perspective. This work initially received mixed ratings. The raised concerns by the reviewers are gradually solved by the authors and finally, all reviewers are positive towards the current form. The AC has checked all files, and stands on the reviewers' side. The authors shall incorporate related results and discussions in the camera-ready version.